



# Spatial variability of northern Iberian rainfall stable isotope values: Investigating climatic controls on daily and monthly timescales

Ana Moreno[1], Miguel Iglesias[2], Cesar Azorin-Molina[3], Carlos Pérez-Mejias[4,1], Miguel Bartolomé[5,6],

Carlos Sancho[Ŧ], Heather Stoll[7], Isabel Cacho[8], Jaime Frigola[8], Cinta Osácar[5], Arsenio Muñoz[5], Antonio Delgado-Huertas[9], Ileana Blade[10] and Françoise Vimeux[11,12]

[1]Department of Geoenvironmental Processes and Global Change, Pyrenean Institute of Ecology – CSIC, Avda. Montañana 1005, 50059 Zaragoza, Spain

[2]Department of Geology, University of Oviedo, C/Arias de Velasco, s/nº 33005 Oviedo, Spain

[3]Centro de Investigaciones sobre Desertificación, Consejo Superior de Investigaciones Científicas (CIDE-CSIC), Moncada 46113, Valencia, Spain

[4]Institute of Global Environmental Change, Xi'an Jiaotong University, Xi'an, 710049, China

[5]Earth Sciences Department, University of Zaragoza, C/Pedro Cerbuna 12, 50009 Zaragoza, Spain

[6]Departamento de Geología. Museo Nacional de Ciencias Naturales – CSIC, C/José Gutiérrez Abascal 2, 28006, Madrid, Spain

[7]Geological Institute, NO G59, Department of Earth Sciences, Sonneggstrasse 5, ETH, 8092 Zurich, Switzerland

[8]CRG Marine Geosciences, Department de Dinàmica de la Terra i l'Oceà, Facultat de Ciències de la Terra, Universitat de Barcelona, C/Martí i Franqués, s/nº, 08028 Barcelona, Spain

[9]Stable Isotope Biogeochemistry Laboratory, IACT-CSIC, Avda. de Las Palmeras nº 4, 18100, Armilla (Granada) Spain

[10]Group of Meteorology, Department of Applied Physics, Faculty of Physics, University of Barcelona, Martí i Franqués,1, 08028 Barcelona, Spain

[11]HydroSciences Montpellier (HSM), UMR 5569 (UM, CNRS, IRD), 34095 Montpellier, France.

[12]Institut Pierre Simon Laplace (IPSL), Laboratoire des Sciences de Climat et de l'Environnement (LSCE), UMR 8212 (CEA, CNRS, UVSQ), 91191 Gif-sur-Yvette, France.

[Ŧ] Deceased, February 2019

*Correspondence to*: Cesar Azorin-Molina (cesar.azorin@uv.es)

**Abstract.** This article presents for the first time a large dataset of rainfall isotopic measurements ($\delta^{18}O_p$ and $\delta^2H_p$) sampled every day or every two days from seven sites in a west-to-east transect across northern Spain for 2010-2017. The main aim of this study is to: (1) characterize rainfall isotopic variability in northern Spain at daily and monthly time scales, and (2) assess the principal influencing factors determining rainfall isotopic variability. This comprehensive spatio-temporal approach allows exploring the role of air mass source in determining the isotopic composition of rainfall in northern Iberia





by using back-trajectories; Atlantic fronts are found to be the dominant source of northern Iberia rain events studied. The relative role of air temperature and rainfall amount in determining the stable isotope composition of precipitation changes along the west-to-east transect. Air temperature appears to be the most significant influence on $\delta^{18}O_p$ at daily and monthly time scales with the highest air temperature-$\delta^{18}O_p$ dependency found for the Pyrenean station while a few sites in the transect show a significant negative correlation with precipitation amount. Distance from the coast, site elevation, and moisture source region (Atlantic versus Mediterranean) also significantly modulate the $\delta^{18}O_p$ values and ranges but the type of precipitation (convective vs frontal rainfall) plays a key control, with convective rainfall associated with higher $\delta^{18}O_p$ values. This dataset of the rainfall isotopic composition represents another step forward towards developing a more detailed, mechanistic framework for interpreting stable isotopes in rainfall as a palaeoclimate and hydrological tracer.

## 1 Introduction

The oxygen isotopic composition of rainfall ($\delta^{18}O_p$) is often considered as the dominant influence on the isotopic composition of terrestrial archives (ice cores, speleothems or authigenic lacustrine carbonates) used to reconstruct past climate (e.g., Leng, 2006). However, few palaeoclimate reconstructions are supported by an in-depth understanding of the regional climatic controls on modern precipitation $\delta^{18}O$ ($\delta^{18}O_p$) (e.g. Treble et al., 2005),. As a consequence, paleoclimate proxies are often interpreted without a clear knowledge of the processes involved in modulating $\delta^{18}O_p$ at a particular region (López-Blanco et al., 2016; Moreno et al., 2017). It has long been established that $\delta^{18}O_p$ is an integrated product of air masses history , modulated by specific prevailing meteorological conditions (air temperature and amount of precipitation for example) (Craig, 1961; Dansgaard, 1964). This results in different dominant factors controlling $\delta^{18}O_p$ variability depending on the site location, i.e., latitude, continentality, elevation, seasonal distribution, local air temperature and amount and source of precipitation (Rozanski et al., 1993). A detailed study of current $\delta^{18}O_p$ values and their variability in a given region is mandatory if one wishes to reconstruct past climate changes using $\delta^{18}O$ in regional climate archives (Lachniet, 2009).

Long rainfall isotopic time series allow for comparison of the $\delta^{18}O_p$ signal with meteorological variables and calibration of proxy records. Unfortunately such long-term observational studies are scarce, and thus, only a few, although outstanding, examples of studies examining factors controlling $\delta^{18}O_p$ are available for continental Europe (Field, 2010; Genty et al., 2014; Tyler et al., 2016). The application of results obtained in other European regions mostly under the influence of rainfall with Atlantic origin (e.g., Baldini et al., 2010) are not valid for the Iberian Peninsula (IP) where three major precipitation regimes coexist (Millán et al., 2005) and where a potential for paleoclimate reconstructions exists through speleothems analyses. Previous studies have shown that the spatial distribution of $\delta^{18}O_p$ and $\delta^2H_p$ at a monthly time scale in Spain can be explained by a simple multiple regression model, based on two geographic factors: latitude and elevation (Díaz-Tejeiro et al., 2013, 2007). However, these models do not reproduce the observed stable isotope composition of precipitation with a detailed spatial resolution. The well-known complex topography and varied weather regimes of Spain (AEMET and Instituto de


Meteorologia de Portugal, 2011; Martin-Vide and Olcina-Cantos, 2001) require further detailed and highly spatially-resolved studies.

A major advance in understanding the controls on $\delta^{18}O_p$ has been the proliferation of studies using daily-scale monitoring to address the mechanisms behind isotopic signatures at daily timescales (Baldini et al., 2010; Fischer and Baldini, 2011), incorporating the complexity associated to the type of rainfall, for example (Aggarwal et al., 2016)Regrettably, the scarcity of *Global Network of Isotopes in Precipitation* (hereafter GNIP) sites in Iberia, particularly those recovering data at a daily scale, prevents a broader regional study of climate controls on $\delta^{18}O_p$ values. In the IP, only one study has analysed $\delta^{18}O_p$

variability at a daily basis covering a short 3-year period (2000-2002) (Araguás-Araguás and Diaz Teijeiro, 2005) and, more recently, a 3-year monitoring survey focused on the Iberian Range (Molinos, Teruel, NE Spain) (Moreno et al., 2014). That study revealed the importance of the *source effect* on $\delta^{18}O_p$ values, due to the alternating influence of two air masses trajectories with different isotopic ranges; basically, Atlantic fronts with more negative $\delta^{18}O_p$ values (from the west) and Mediterranean convective storms with more positive values (from the east) air trajectories (Moreno et al., 2014).

Additionally, another recent study based on back trajectories emphasized the role of recycled moisture uptake within the IP in the final values of $\delta^{18}O_p$ in Central Spain (Eagle Cave) (Krklec and Domínguez-Villar, 2014). In addition, to date, the majority of empirical studies of the meteorological controls over $\delta^{18}O_P$ rely upon event scale, daily, or monthly time series from individual locations (Smith et al., 2016), which raises concerns about the spatial representativeness of the resulting statistical models and the mechanisms behind those relationships.

In this paper, we propose an alternative approach by analysing daily and monthly patterns of $\delta^{18}O_P$ from *multiple* stations across northern IP, and the Balearic Islands, following a west-to-east transect (850-km in straight line) that extends from an area dominated by a typical Atlantic climate to fully Mediterranean sites. The overall aim is to characterize and quantify the dominant factors modulating $\delta^{18}O_p$ variations in time (daily and monthly) and space, to determine the causes of precipitation isotopic variations regionally. Additionally, this study will serve to improve the interpretation of oxygen isotope paleo-

records from the region that depend on $\delta^{18}O_p$ (Bartolomé et al., 2015; Domínguez-Villar et al., 2017; López-Blanco et al., 2016; Pérez-Mejías et al., 2019; Sancho et al., 2018, 2015).

**2 Prevailing climate regime and site description**

Our study compares for the first time rainfall isotopic values and meteorological variables (temperature, precipitation, moisture sources and type of rainfall) at seven sites in northern Iberia and Balearic Islands, covering an 850-km long west-to-east transect from an area under typical Atlantic (Oviedo and El Pindal) to fully Mediterranean (Mallorca Island and Barcelona) climate. The west-to-east transect is completed with three additional sites in a transitional zone: two from the Iberian Range (Molinos and Ortigosa de Cameros) and one from the Pyrenees (Borrastre) (Figure 1a). At those seven

locations rainfall was sampled daily covering different time periods except at El Pindal where rain was collected every 48h



(Table 1). Borrastre record is, to our knowledge, the most comprehensive dataset of daily $\delta^{18}O_p$ for Spain in terms of both the time span covered (2011-2016) and number of samples (380 days) (Table S1).

In north-western and north-central Iberia, precipitation is mainly controlled by the presence of westerly winds and the passage of Atlantic fronts, mainly during November-April (Martín-Vide and Olcina Cantos, 2001). During the rest of the
year, the subtropical Azores high-pressure system shifts northward, favouring stable conditions by blocking the westerly circulation and moisture inflow (Archer and Caldeira, 2008), thus reducing precipitation. This wet winter/dry summer regime is quite different from that in the north-eastern Mediterranean region, where winters are generally dry (foehn effect) whereas warm season precipitation (from late spring to early autumn) is dominated by convective storms and also easterly advections over the Mediterranean Sea (backdoor cold fronts) (Millán et al., 2005). These local to mesoscale storms are primarily
associated with frequent and persistent sea breezes (Azorin-Molina et al., 2011) which bring warm and moist air masses from the Mediterranean sea inland (Azorin-Molina et al., 2009). During the summer season, this is typically the only source of precipitation in the north-eastern area, bringing an average of 100-125 mm yearly (Millán et al., 2005). Backdoor cold fronts from the Mediterranean Sea are sporadic events occurring in autumn (secondarily in winter-spring), but they cause heavy precipitation and flooding (Llasat et al., 2007). Figure 1B summarizes these three major precipitation regimes defined by
Millán et al. (2005): (i) Atlantic frontal systems (westerly winds), (ii) convective–orographic storms, and (iii) Backdoor cold fronts from the Mediterranean Sea (easterly winds).

Winter precipitation in large parts of the IP is strongly influenced by the North Atlantic Oscillation (NAO) at annual and interannual scales: higher precipitation occurs when the NAO index (NAOI) is negative and the storm tracks are shifted southwards, more directly influencing the IP (Trigo et al., 2002). Lower correlation values ($r$ =-0.1-0.4) between the NAOI
and winter rainfall, however, are observed in our region of interest, the northern IP (Goodess and Jones, 2002), which encompasses both the wet western regions and the dry Mediterranean in the northeast. For the latter region, a significant relationship with the Western Mediterranean Oscillation (WeMO) in spring and autumn is attributed to fluctuations of warm moist inflow air from the east and its influence on Mediterranean cyclogenesis (Martin-Vide and Lopez-Bustins, 2006).

The rainfall influencing the seven stations included in the studied transect originates in two dominant source regions: the
tropical-subtropical North Atlantic and the Western Mediterranean (Gimeno et al., 2010). Below, the four regions across which the seven stations are distributed are described in terms of their climatology. Regional meteorological data are provided in Figure 4A.

The Cantabrian coast. The site of El Pindal in the Cantabrian coast (Figure 1A) is characterized by a typical oceanic climate with mild summers and winters (Cfb, following Köppen and Geiger – KGC- classification) due to the proximity to the coast.
Rainfall mainly occurs in late autumn and early winter with a minimum in summer (Figure 4A), and are associated with Atlantic frontal systems (westerly winds). Additionally, rainfall samples from Oviedo (climate characteristics similar to those at El Pindal) were collected and are also included in this study.





The Iberian Range. Ortigosa de Cameros is located in the Encinedo Mountain area in the westernmost sector of the Cameros Range (Iberian Range, Figure 1A) and is dominated by a continental Mediterranean climate (Dsb, following KGC

classification). Rainfall mostly occurs in autumn and spring, with some convective-orographic storms in summer (climograph in Figure 4A). Also located in the Iberian Range and at similar elevation but further east, in the Maestrazgo basin, the Molinos site is characterized by similar climate (Dsb in KGC classification), with a highly pronounced seasonality and precipitation occurring mainly in spring and in autumn (Figure 4A).

The Pyrenees. Borrastre village is located in the Central Pyrenees (Figure 1A) and is influenced by a transitional climate

Mediterranean-Oceanic (Csb in KGC classification), with precipitation occurring mainly in spring and, to a lesser extent, in autumn (Figure 4A), exhibiting a mix of the three Atlantic, Mediterranean and convective precipitation regimes.

The Mediterranean. The typical Mediterranean climate (Csa in KGC classification) is represented by the Manacor and Porto Cristo localities in the Mallorca island and by Barcelona (Figure 1A). Precipitation is mostly distributed from October to April (Figure 4A) mostly associated with backdoor cold fronts from the Mediterranean Sea (easterly winds) as the influence

of Atlantic precipitation is weakened over this area.

## 3 Analytical and statistical methods

### 3.1 Sampling


Rainwater was collected using a similar procedure to that recommended by the International Atomic Energy Agency (IAEA) for daily sampling (http://www-naweb.iaea.org/napc/ih/IHS_resources_gnip.html) for six of the seven stations (Oviedo, Ortigosa de Cameros, Molinos, Borrastre, Mallorca and Barcelona). Thus, precipitation events greater or equal to 1 mm were sampled from a rain gauge which allows measuring the amount of rain fallen and sample it manually taking out the water

from the rain gauge with a syringe. The collected water was then homogenized and filtered at the time of sampling, later a 5ml aliquot was stored in polypropylene tubes sealed with screwcup without air inside and kept cold in a refrigerator until isotopic analysis. Rainfall samples were collected at the end of each precipitation event, immediately afterwards whenever possible or after a few hours, with the total event precipitation homogenized. At El Pindal site the procedure was different: rainfall was collected every 48h for several months (November 2006 to April 2009) using an automated sampler

(Table 1) located on the roof of the San Emeterio lighthouse located <10 m from the modern sea cliff and 200 m from the cave. Thus, since the samples were automatically collected and remained in the lighthouse for several days, a film of paraffin oil was used to prevent evaporation.

The observation staff in charge of each location collected a sample directly following every rainfall event, except in El Pindal that the system was automatic and in Mallorca, where several events were missed during the first two years of the

collection period, preventing the calculation of monthly averages for some intervals (monthly and annual averages and standard deviations in Table 2). In addition, seven rainfall events were collected at two different localities in Mallorca





(Manacor and Porto Cristo) obtaining similar $\delta^{18}O_p$ results. For those events, a weighted-average value using the two localities was calculated (see Table S1). Thus, 47 rainfall samples were collected from Oviedo manually in 2015. In Ortigosa de Cameros, rainfall was manually collected daily between September 2010 and December 2014 by the staff (guides) of the
La Viña and La Paz show caves, with an interruption from December 2012 – January 2014. In Molinos, rainfall was manually collected by the staff of the Grutas de Cristal cave every day for just over five years (March 2010-May 2015). The first 2.5 years rainfall data from that survey was previously published (Moreno et al., 2014; Pérez-Mejías et al., 2018). In Borrastre, rainfall was manually collected daily using a Hellman rain gauge daily from April 2011 to May 2016 (380 events). In Barcelona, rainfall samples were obtained from the weather station on the roof of the School of Physics of the University
of Barcelona using a standard rain gauge.

## 3.2 Analytical methods

The isotopic composition of oxygen and hydrogen in rainfall samples, expressed as $\delta^{18}O$ and $\delta^{2}H$, reported in ‰ relative to Vienna Standard Mean Ocean Water (VSMOW). Molinos, Borrastre and most of Ortigosa de Cameros samples (143 samples) were analysed using a Finningan Delta Plus XL mass spectrometer at the IACT-CSIC in Granada. Water samples
were equilibrated with $CO_2$ for the analysis of $\delta^{18}O$ values (Epstein and Mayeda, 1953), while the hydrogen isotopic ratios were measured on $H_2$ produced by the reaction of 10 μL of water with metallic zinc at 500°C, following the analytical method of Coleman et al. (1982). The analytical error for $\delta^{18}O$ and $\delta^{2}H$ was ±0.1 and ±1 ‰, respectively. The Mallorca and Barcelona samples and the remaining samples from Ortigosa de Cameros (50 samples) were analysed at the Scientific and Technological Centre from the University of Barcelona, $\delta^{2}H$ via TCEA pyrolysis coupled to Thermo Delta Plus XP mass
spectrometer and $\delta^{18}O$ with a MAT 253 Thermofisher spectrometer coupled with a gas bench. The analytical error for $\delta^{18}O$ and $\delta^{2}H$ was ±0.2 and ±1.5 ‰. El Pindal samples were measured at three different laboratories (see Stoll et al., 2015, for more details). Rainfall collected from November 2006 through the end of February 2007 was analysed at the University of Barcelona using the procedure described above. Rainfall collected from June 2007 to May 2008 was analysed in the Marine Biological Laboratories of the University of Oviedo, using equilibration with $CO_2$ on GV Multiflow-Bio unit coupled to a
GV ISOPRIME CF mass spectrometer. Rainfall collected from June 2008 to April 2009 and samples from 2015 were analysed using equilibration with $CO_2$ on Gas Prep unit coupled to a Nu Instruments Horizon mass spectrometer at the University of Oviedo. Uncertainties are ±0.1‰ (1s) for $\delta^{18}O$ and ±1 %.for $\delta^{2}H$, based on replicate analyses. Unfortunately, no comparison was made between the different involved laboratories and thus the study does not account for possible offsets between them.
Additionally, 18 samples of potentially evaporated water with abnormally high values in $\delta^{18}O_p$ - and that occurred in summer months when maximum daily air temperatures exceed 30°C - were classified as outliers and removed from the database. These 18 samples (Table S1) were from Ortigosa de Cameros (4 samples), Borrastre (6 samples) and Molinos (8 samples). Partial evaporation of falling rain-droplets is an alternative interpretation of the high $\delta^{18}Op$ values of these samples.



### 3.3 Meteorological data


Meteorological data to investigate the statistical relationship between isotopic values (at daily and monthly time scales) and main climate variables (air temperature and precipitation) were obtained from the closest meteorological stations over the sampling periods as indicated in Table 1. For Oviedo, meteorological data are obtained from Oviedo AEMET station and for

El Pindal (120 km from Oviedo; 70 km from Santander) since there was not good data from nearby stations, we decided to use ERA-Interim re-analysis of the European Center for Medium-range Weather Forecasts (ECMWF) that provides gridded weather data (Berrisford et al., 2009). For Ortigosa site, meteorological data are obtained from Villoslada de Cameros meteorological station (http://www.larioja.org/ emergencias-112/es/meteorologia), at 6.5 km from the rainfall collection site. The Borrastre sampling site has its own meteorological station (http://borrastre.dyndns.org/MeteoBorrastre) (Table 1),

except for the first 22 events that were derived from ERA-Interim since the station was not yet operative. Finally, for Mallorca we used data from Sant Llorenç station (8 km) while Barcelona meteorological data are obtained from Zona Universitaria station (www.meteo4u.com).

### 3.4 Statistical analyses


Prior to conducting correlation analysis at daily scale, we removed the seasonal component of the variables by subtracting their monthly averages to avoid sympathetic seasonal correlations (e.g. (Kawale et al., 2011; Rozanski et al., 1993) (Table 3A). To establish correlations at monthly scale with meteorological variables (Table 3B), $\delta^{18}O_p$ monthly averages weighted by the amount of precipitation were calculated using the following formula (Figure 4B):

$\delta^{18}O_{monthly} = ((Q1 \times \delta^{18}O_1) + (Q2 \times \delta^{18}O_2) ... (Q_i \times \delta^{18}O_i)) / (Q_1 + Q_2 + ... Q_i)$ [1]

with Q = rainfall quantities for the day i (in mm). Daily values were not averaged since there was only one rainfall sample per day resulting from the homogenization of all the event samples of that day. Spearman's rank correlation analysis, a non-parametric alternative to Pearson correlation analysis was preferred to account for non-linear relation, with r as the correlation coefficient (PAST software, Hammer et al, 2001) (Table 3).The analyses were conducted at daily (Table 3A) and

monthly (Table 3B) time scales. Bonferroni test was applied to prevent data from incorrectly appearing to be statistically significant by making an adjustment during comparison testing. Additionally, to integrate the temperature effect and the amount effect, a multiple regression model for $\delta^{18}O$ was carried out using PAST software for every studied site (Table 3C). Backward-trajectory analysis was performed using the Hybrid Single-Particle Lagrangian Integrated Trajectory (HYSPLIT) Model (Version 4.8) (Draxler and Rolph, 2010) and following a similar methodology to Baldini et al. (2010) over a 120 hours

lifetime (Figure 5) (24 hours trajectories were also calculated, Fig. S1). All points produced by the HYSPLIT model every hour (120 points) were used to generate a vector representing the origin and mean trajectory of the rainfall collected. Once





all the 120 vectors were produced, they were averaged, and one unique vector was assigned to each rainfall event. After that, all the averaged vectors associated with each different location studied, are presented in a compass rose using 10° intervals, together with $\delta^{18}O$ values and rainfall amount of each event (mm) provided by the closest weather stations to each analyzed
location (Figure 5).

Lastly, to better explore the role of the type of precipitation in controlling the isotopic composition of rainfall across northern Iberia, we applied a disaggregation procedure of precipitation series following the same methodology described in Millan et al. (2005). This novel method classifies each precipitation event on the basis of its characteristics, distinguishing between three categories (Figure 1B, Table 4): (i) frontal systems associated with passing cold fronts from the west, (ii) convective-
orographic storms driven by differential heating, sea breezes and local winds (Azorin-Molina et al., 2009) and (iii) easterly advection from the Mediterranean Sea (backdoor cold fronts). The Kruskal-Wallis H test (sometimes also called the "one-way ANOVA on ranks") is a rank-based nonparametric test (Hammer et al., 2001) that was applied to the three rainfall categories to determine if there were statistically significant differences on their $\delta^{18}O_p$ distributions (Table 5).

**4 Results**

**4.1 Daily scale rainfall isotopic variability in northern Iberia**

The rainfall samples for the studied stations at a daily scale, define local meteoric water lines (LMWL) that are roughly
parallel for all sites with similar offset from the Global Meteoric Water Line (GMWL, $\delta^2H = 8*\delta^{18}O + 10$) (Figure 2). All the slopes and the intercepts are lower than the GMWL, with slopes ranging from 6.9 to 7.2 and intercepts from 1.05 to 6.4 (Figure 2).

Despite the fact that the data were not collected for the same period of time at each station, the different daily series of $\delta^{18}O_p$ at all stations are presented together versus time (Figure 3). From 2010 to 2017, daily $\delta^{18}O_p$ clearly show lower values in
winter and higher (sometimes positive) values in summer at all stations (Figure 3). Yet some summer rainy episodes (e.g., the 25 June 2014 event in Borrastre or the 18 June 2016 one in Barcelona) exhibit values typical of winter after raining several days or after an intense rainfall event (41.6 mm in Borrastre or 18 mm in Barcelona).

Also evident in Figure 3 is the synchronicity among stations for specific events. A good example is the episode of 16-18[th] November 2013 (inset in Figure 3) when very negative values were reached at Molinos (black line), Borrastre (green line)
and Mallorca (red line). This period was characterized by intense widespread rain – eg. 43 mm in Mallorca and 36 mm in Molinos (Table S1). At the three sites, this period was among the rainiest of our record with some of the lowest $\delta^{18}O_p$ values recorded.

**4.2 Monthly scale rainfall isotopic variability in northern Iberia**






Seasonality in $\delta^{18}O_p$ in northern Iberia is further explored in Figure 4B (data in Table 2, Table S2). Al stations exhibit a clear seasonal pattern in temperature with a peak in July/August and minimum values in December/January. The seasonal signal in $\delta^{18}O_p$ roughly follows this pattern with peak $\delta^{18}O_p$ values in summer and minimum $\delta^{18}O_p$ in winter. It is worth noting that precipitation exhibits a bi-modal pattern which is not reflected in $\delta^{18}O_p$. The average seasonal differences between July-

August and minimum $\delta^{18}O_p$ in January-February are quite large: 5.8‰ at Borrastre, 4.6‰ at Ortigosa de Cameros, 6.2‰ at Molinos and about 4‰ at Mallorca-Barcelona. Interestingly, the Oviedo-El Pindal samples reveal a very different pattern, with a marked reduction in seasonality compared to the other sites (2 ‰ $\delta^{18}O_p$ difference between winter and summer) (Figure 4B).

**5 Discussion**

This discussion section is focused on analysing the main factors controlling $\delta^{18}O_p$ in the studied transect in northern Spain at daily and monthly time scales. Sect. 5.1 is dedicated to the influence of geographical parameters, such as distance to coast or elevation of the studied sites. Sect. 5.2 deals with the role of meteorological parameters, in particular, local air temperature

and precipitation amount. Sect. 5.3 investigates the role of moisture origin on $\delta^{18}O_p$ variability while Sect. 5.4 explores the role of rainfall type (convective, frontal) in determining $\delta^{18}O_p$.

**5.1 Geographical controls on rainfall isotopic variability**

The combination of the various isotope effects results in consistent and spatially coherent variation in $\delta^{18}O_p$ values that are primarily related to latitude, elevation, moisture source and air masses history (Rozanski et al., 1993; Bowen, 2008). The LMWLs determined with daily data for each of the studied sites reveal a broadly similar regional signal and are consistent with previous studies using GNIP data from southern France (Genty et al., 2014), even considering that study is made with monthly $\delta^{18}O_p$ data. The slopes obtained are slightly lower in our study compared to a previous analysis from the IP

(Araguás-Araguás and Diaz Teijeiro, 2005) where the sampling period only covered the rainy season (October to March). The comparison of monthly averaged $\delta^{18}O_p$ values in the studied stations allows an assessment of the relative importance of geographical factors in the observed patterns (Table 2, Figure 4B). Ortigosa de Cameros, Molinos and Borrastre stations show monthly $\delta^{18}O_p$ values quite similar and, normally, more negative than Oviedo, El Pindal, Barcelona and Mallorca sites. This pattern is particularly clear for autumn values (see monthly averaged $\delta^{18}O_p$ values from September to December in

Table2 along the west-to-east transect). The similarity found among the sites located at opposite ends of the transect (Oviedo and El Pindal compared to Barcelona and Mallorca) and, presumably, influenced by different air masses with different isotopic composition in the initial water vapor (i.e., Atlantic vs Mediterranean) as previously described by LeGrande and



Schmidt (2006) in their global study of oxygen isotopic composition in seawater, is not what we expected. This pattern may be explained by two processes.

First, the fact that Oviedo and El Pindal rainfall samples show enriched $\delta^{18}O_p$ values (Table 2, Figure 4B) is consistent with their location in the Cantabrian coast, very close to the Atlantic Ocean, with oceanic climatological conditions characterized by high mean temperatures (Table 1). Thus, Oviedo (and El Pindal) are the stations that receive the first precipitation produced by Atlantic air masses; therefore they are the stations in the transect least affected by the "continental effect": when clouds move inland from the Atlantic Ocean and become gradually isotopically depleted due to progressive rainout

(Dansgaard, 1964). Thus, as we follow the typical movement of an Atlantic front, from west to east, we find progressively more negative winter $\delta^{18}O_p$ values (considering an average of January-February-March, Table 2) going from El Pindal (-6.0‰) to Ortigosa de Cameros (-8.1‰), to Borrastre (-9.8‰) and, finally, to Molinos (-10.0‰). This pattern is not so evident in other seasons where the entrance of Atlantic fronts is not the only synoptic pattern that generates rainfall in the transect. However, the large observed differences cannot be explained only by this effect that accounts for a very small

variation (about 0.002‰ per km in Europe as described in Rozanski et al., 1993).

A second factor to explain why Mallorca and Barcelona rainfall samples display the least negative $\delta^{18}O_p$ values in the transect, is the influence of air masses derived from the Mediterranean Sea. Initial water vapour $\delta^{18}O$ values in the Mediterranean Sea are typically more positive (0.5 – 1 ‰) than Atlantic Ocean water vapour due to enhanced evaporation in within the semi-enclosed Mediterranean basin (LeGrande and Schmidt, 2006). An additional effect, probably more important

than 0.5 - 1 ‰ of difference, is the higher annual mean air temperature at those stations (Mallorca and Barcelona together with Oviedo and El Pindal) compare to the other ones (Table 1). The effect of temperature to produce the less negative $\delta^{18}O_p$ values recorded will be explained below (Sect. 5.2).

In addition, the three stations with more negative monthly $\delta^{18}O_p$ values (Ortigosa de Cameros, Borrastre and Molinos) are at higher elevation than the other stations that are all located close to sea-level. Therefore, the "elevation effect" (Siegenthaler

and Oeschger, 1980) likely also plays a role in explaining the more negative $\delta^{18}O_p$ values at those stations. Considering the $\delta^{18}O_p$ annual averages (Table 2) there is a difference of 2.3‰ between Molinos (1040 m asl) and Mallorca (90 m asl). Based on the difference of elevation, the vertical isotopic gradient observed is -0.24‰ per 100 m of elevation. This result is consistent with previous studies in other mountain ranges such as the Alps, where an altitudinal gradient of -0.2 to - 0.3‰ per 100 m of elevation was observed (Ambach et al., 1968; Siegenthaler and Oeschger, 1980). However, in spite the

difference in elevation, we need to consider that the sites are very distant and separated by the Mediterranean Sea. Therefore, the altitude cannot be the only parameter controlling the differences between the studied sites.

Finally, the geographical factors reviewed in this section (distance to the coast or continental effect, elevation effect, and $\delta^{18}O$ composition of the sea waters) exert a small direct influence on the observed spatial distribution of rainfall $\delta^{18}O_p$ at the studied sites but contribute to the effects of other, controlling factors: air temperature, rainfall amount, air mass trajectory

and rainfall type, which will be described in following Sect. 5.2, 5.3 and 5.4.





## 5.2 The influence of air temperature and rain amount on the spatial distribution of rainfall $\delta^{18}O_p$ values at daily and monthly time scales

Spearman's rank correlation analysis (Table 3) reveals that $\delta^{18}O_p$ does not correlate with air temperature or amount of precipitation in a similar way at each station, neither at daily or monthly scales, thus reinforcing the need for conducting such studies on a local basis particularly when conducting paleoclimate reconstructions (Leng, 2006). Air temperature appears as the most robust influence across the west-to-east transect, with low but statistically significant correlations (daily scale) with $\delta^{18}O_p$ at all sites (red numbers in Table 3A) except Oviedo and Barcelona, most likely due to the low number of daily

samples (n=39 and n=53, respectively). The coefficient of correlation among $\delta^{18}O_p$ daily values and air temperature is highly variably from west to east: El Pindal ($r_s$ = 0.34; $p$ = 0.0012), Ortigosa de Cameros ($r_s$ = 0.25; $p$ = 0.001), Molinos ($r_s$ = 0.42; $p$ = 2.00E-11), Borrastre ($r_s$ = 0.29; $p$ = 6.33E-08) and Mallorca ($r_s$ = 0.35; $p$ = 0.0013) (Table 3A). Regarding monthly values, air temperature is significantly correlated with $\delta^{18}O_p$ values at eastern stations, with the highest coefficients associated with higher altitude sites (e.g., in Molinos with $r_s$ = 0.76 and $p$=3.36E-10 or in Borrastre with $r_s$ = 0.61 and

$p$=1.44E-05) (Table 3B).

The dependence of $\delta^{18}O_p$ on air temperature has been extensively studied, yielding an *average* slope for mid-latitude continental stations of 0.58‰/°C (Rozanski et al., 1993). However, that value is highly variable in time and space. The strongest air temperature-$\delta^{18}O_p$ relationship based on daily data is found at the Pyrenean station, Borrastre site with 0.4‰/°C and the weakest at Oviedo+El Pindal (0.2‰/°C). The other three stations, Ortigosa de Cameros (0.3‰/°C), Molinos

(0.4‰/°C) and Mallorca+Barcelona (0.3‰/°C), show similar intermediate values. Compared to other areas, such as the Alps with temperature-$\delta^{18}O_p$ gradients of 0.5 to 0.7 ‰ per °C, the results presented above indicate that, although important, air temperature only explains between 20 and 40 % of the observed $\delta^{18}O_p$ variability and is, therefore, not the only control.

The amount effect is dominant in tropical regions where deep vertical convection is common although it may also occur in the extratropics (Bar-Matthews et al., 2003; Treble et al., 2005b). In the studied transect, at the daily scale, the strongest

correlation with amount of precipitation is observed in Barcelona ($r_s$ = -0.35; $p$=0.029) (Table 3A). Besides, there is a significant correlation at the two sites of the Iberian Range ($r_s$ = -0.32; $p$=1.05E-05 in Ortigosa and $r_s$ = -0.19; $p$=0.005 in Molinos). Interestingly, the westernmost stations (El Pindal and Oviedo) do not show a significant $\delta^{18}O_p$–precipitation correlation at the daily or monthly scale. This lack of correlation in the Atlantic sites (El Pindal and Oviedo) contrasts with a previous study carried out in northern Spain and also characterized by an Atlantic climate (Matienzo depression) where there is found a significant $\delta^{18}O_p$–precipitation monthly correlation ($r$ = -0.51; $p$ < 0.01) (Smith et al., 2016). In our study, the

$\delta^{18}O_p$–precipitation correlation at monthly scale is only significant in Molinos, in the Iberian Range ($r_s$ = - 0.4; $p$=0.018) while no correlation is observed in the other sites (Table 3B).



To further assess the relative role of temperature and amount effects, a multiple regression model for $\delta^{18}O_p$ was carried out for the seven studied sites in which the temperature effect exerted a clear dominant control (Table 3C. Still, both influences

together account for less than 20 % of the variability of $\delta^{18}O_p$ in the study transect. Since the origin of rainfall and type of rainfall (i.e., convective vs. frontal) is also spatially dependent in northern Iberia, these variables and their influence on the observed $\delta^{18}O_p$ variability are investigated in Sect. 5.3 and 5.4 below.

**5.3 The role of the *source effect* in modulating northern Iberian Peninsula $\delta^{18}O_p$**


The source effect describes how air masses derived from different moisture sources have distinct $\delta^{18}O_p$ values (e.g., Friedman, 2002). The source effect results from varying air mass histories, different conditions of the moisture source (temperature, relative humidity and wind speed) and regional differences in the $\delta^{18}O$ of the surface ocean (LeGrande and Schmidt, 2006). In the case of northern IP, it is necessary to consider the effect of both the Atlantic Ocean and Mediterranean

Sea as important sources of atmospheric moisture (Gimeno et al., 2010) whose relative influence on regional IP $\delta^{18}O_p$ could be very different because of the complex regional topography of the area. General $\delta^{18}O$ values of seawater reconstructions (LeGrande and Schmidt, 2006) indicate different values for the Atlantic Ocean and the Mediterranean Sea due to temperature and salinity differences. Source $\delta^{18}O$ values range from 1 to 1.5‰ in the subtropical Atlantic to 2‰ in the Mediterranean (Schmidt et al., 1999). Although these differences (about 0.5 - 1 ‰) are small, since they are further

modulated by the air mass history, we expect to see a change in the relative influence of moisture source on $\delta^{18}O_p$ along the west-to-east transect.

Evaluation of monthly $\delta^{18}O_p$ patterns represented in Figure 4B reveals more muted seasonality (2‰) at Oviedo-El Pindal sites compared to other stations in the transect (> 4‰). The seasonal difference from winter to summer in Oviedo is similar to the values published by Genty et al., (2014) for stations in southern France (e. g., 2.1 ‰ in Villars with only Atlantic

influence and 3.6 ‰ in Orgnac with Atlantic and Mediterranean influence). The explanation for the weak seasonality in the Oviedo $\delta^{18}O_p$ signal and the similarity with Villars station could be related to the precipitation type (Sect. 5.4) and the geographic origin. Oviedo and Villars stations are characterized by a relatively constant source of precipitation through the year derived from Atlantic fronts and no dry season (Figure 4A). This is in clear contrast to the other stations which are characterized by a more hybrid Atlantic/Mediterranean climate (e.g., Orgnac, Genty et al., 2014). Particularly, in Barcelona

and Mallorca the seasonal difference in $\delta^{18}O_p$ monthly values is high (6‰) (Figure 4B). At these two stations, the influence of different rainfall sources (Atlantic and Mediterranean) with distinct isotopic values as demonstrated by a global study of $\delta^{18}O$ values in surface oceans (LeGrande and Schmidt, 2006) and different air masses histories may be important to explain the high variability. These influences are further evaluated using back trajectory analysis.



To evaluate the role of moisture source in determining $\delta^{18}O_p$ values at a daily scale in northern Iberia and Balearic islands,
back trajectories were calculated for all the rainy days and subsequently averaged into wind rose diagrams, following representation applied in previous studies (Smith et al., 2016), for three stations along our northern Iberia transect: Oviedo and Mallorca, the two extreme locations of the studied transect, and Borrastre, situated at an intermediate location, representing a total number of 519 events (Figure 5). To facilitate statistical comparison of the mean trajectory paths and moisture uptake regions with the oxygen isotope signature of sampled rain events, the vector angle between every site
(Oviedo-Borrastre-Mallorca) and each hourly position along 120-h back trajectories (at 700 and 850 hPa) for each event was estimated, following the methodology presented in Baldini et al. (2010) (Figure 5). Once all the vectors were produced for each sampled event, they were averaged, and presented in a compass rose using 10° intervals, together with $\delta^{18}O_p$ values and rainfall amount of each daily sample (mm) provided by weather stations closed to each location analyzed (Figure 5).

This analysis reveals the dominance of western trajectories in the three studied sites, with very few episodes associated with
other directions (Figure 5). Only some episodes from SW (e.g., Borrastre) or SE (e.g., Mallorca) trajectories are found and, interestingly, they have distinct $\delta^{18}O_p$ value (see below). This low, almost negligible, presence of trajectories associated with Mediterranean air mass advections, does not inhibit the possibility of a moisture uptake over the Mediterranean or moisture recycling with altitude in the mountain region surrounding Borrastre since meteorological processes connected to convection (e.g., orographic, dynamic, thermal) can produce moisture uptake in less than 6h (Romero et al., 2000, 1997; Tudurí and
Ramis, 1997) and may not be well-captured in the back trajectory analyses, which are computed for the previous 120 hours (see Methods). Therefore, convection processes, that may be associated with easterly trajectories, are under-represented in this methodology (see 24 hours analyses in Figure S1 where more trajectories with different origin appear more frequently). Therefore, it is important to note here that this method provides information on the air mass origin (source effect) but not in the moisture uptake regions. In that way, it is clear the dominant WNW trajectory for the three studied stations.

Despite the three sites sharing a common dominant WNW trajectory, they behave quite differently in terms of the associated amount of rainfall and $\delta^{18}O_p$ values. Oviedo (with a temperate oceanic climate - Cfb, Table 1) presents a narrower range of rainfall amounts and $\delta^{18}O_p$ values than at the other two sites, as shown in Figure 5A by the negligible frequency of rainfall amounts above 32 mm (orange) or below 2 mm (purple), while "extreme" events are much more common in Borrastre or Mallorca sites. Similarly, in figure 5B, where the isotopic values for the different trajectories are plotted, Oviedo appears as
the station with more stable $\delta^{18}O_p$ values ($\delta^{18}O_p$ values among -10 and -2‰) compared to the other two. Thus, in Borrastre and Mallorca, $\delta^{18}O_p$ values between -8 and -12‰ (red – green – yellow - dark blue) are only present in northwestern trajectories, while less negative values (- 6 to 2‰) appear in events with SW and SE directions. These results confirm the homogeneity of Atlantic sites (Oviedo, El Pindal) compared to the intermediate (Ortigosa de Cameros, Molinos and Borrastre) stations already described by monthly data in Figure 4.

These results highlight the importance of moisture source in generating the observed $\delta^{18}O_p$ differences along the west-to-east transect in this study. At Borrastre (our mid-transect site) two mean trajectories are distinguished in terms of $\delta^{18}O_p$ values:



northwesterly trajectory associated with more negative $\delta^{18}O_p$ values and southwesterly trajectory associated with less negative values (Figure 5B). Borrastre station is chosen to further evaluate back trajectories for all rainfall events over one whole year (2014, n=126 rainfall events) since the presence of rainfall events where moisture comes from the SW, with
usually less negative $\delta^{18}O_p$ values, is significant compared to, for example, Oviedo station. Thus, one example from every trajectory is presented in Figure 6.

Above 80% of winter trajectories recorded in Borrastre rainfall events originate in the North Atlantic, Artic or inland USA or Canada. They cross the Atlantic Ocean north of Madeira Island and usually enter the IP by the west, next to the Galicia and Portugal border. Those trajectories arriving from the N-NW reach Borrastre site at the Pyrenees almost without crossing the
IP, thus providing the more negative $\delta^{18}O_p$ values (e.g., 7th February, Figure 6A, with $\delta^{18}O_p$ =-6.5‰). On the contrary, those arriving from the W-SW enter via Lisbon and cross central IP providing less negative $\delta^{18}O_p$ values (e.g., 16th January, Figure 6B, with $\delta^{18}O_p$ =-1.2‰). If the trajectory of the air mass travels larger distances over the continent, the contribution of re-evaporated land moisture to the water vapour travelling inland may be significant and thus $\delta^{18}O_p$ values may appear higher, as it has been shown to occur in other regions (Krklec and Domínguez-Villar, 2014). In that case, the progressive rainout
effect may be compensated by the moisture uptake of evaporated (high $\delta^{18}O$) surface water.

During spring, the typical situation of air masses entering from the W alternates with those arriving from the SW, entering at the latitude of the Cape San Vicente and crossing the IP from south to north (e.g., 20th April, Figure 6C; with $\delta^{18}O_p$ =-2.1‰). Some spring trajectories are subject to Mediterranean influence (eg. 20th May; Figure 6D) and are characterized by higher $\delta^{18}O_p$ values ($\delta^{18}O_p$ =-1.3‰). In general, the penetration of subtropical Atlantic air masses, which becomes a very common
situation in summer, results in higher $\delta^{18}O_p$ values (e.g., 6th July, Figure 6E, with $\delta^{18}O_p$ =-2.2‰). Therefore, the less negative $\delta^{18}O_p$ values usually associated with SW trajectories in Borrastre can be explained by (1) the origin in the subtropical Atlantic Ocean with higher $\delta^{18}O_p$ values (1.5 ‰) compared to North Atlantic (0.5 ‰) (LeGrande and Schmidt, 2006) and, (2) the recycling of surface moisture over land incorporating enriched $\delta^{18}O_p$ values from surface waters that have been subject to evaporation over time (Krklec and Domínguez-Villar, 2014).

## 5.4 The influence of rainfall type on isotopes.

The influence of rainfall type on the $\delta^{18}O_p$ is well documented globally, with different $\delta^{18}O_p$ observed depending the type of precipitation (convective showers, frontal, continuous stratiform precipitation, etc.) (Aggarwal et al., 2012). This relationship
is observed in previous studies both at daily or monthly timescales (Aggarwal et al., 2016), with few examples in the Equatorial Indian Ocean (Gat, 1996) and California, USA (Coplen et al., 2015), both indicating that $\delta^{18}O_p$ values were lower when precipitation was dominantly stratiform and higher when it was mostly convective. The main reason to explain this difference lies on the processes of condensation and riming associated with boundary layer moisture which produced higher





isotope ratios in convective rain (Aggarwal et al., 2016). Some studies in the Mediterranean region (Celle-Jeanton et al.,
2001) also directly link the isotopic signature of the precipitation to the prevailing weather conditions during the rainfall
event.

Here we explore how the specific synoptic situation, i.e., rainfall types or rainfall components, influence $\delta^{18}O_p$ values across
the studied transect. Table 4 shows the percentage of rain events associated with each type of precipitation, that were
previously defined following (Millán et al., 2005) and represented in Figure 1B: (i) Atlantic frontal systems (westerly winds),
(ii) convective–orographic storms, and (iii) backdoor cold fronts from the Mediterranean Sea (easterly winds). Backdoor
cold fronts from the Mediterranean Sea are sporadic events occurring in autumn (secondarily in winter-spring), but they
cause heavy precipitation and flooding (Llasat et al., 2007).

The prominence of rainfall associated with Atlantic fronts is evident (above 40% in the seven studied stations). This
percentage decreases eastward, from 68.09/71.29 % in Oviedo/El Pindal to 58.49/40.82 % in Barcelona/Mallorca. A
previous study at a north Iberian site (Matienzo, Cantabria) indicates that approximately 80% of air masses originate in the
North Atlantic, and their movement is associated with westerly frontal systems (Smith et al., 2016). This situation appears to
be true along the studied transect, however for the Mediterranean and Iberian Range sites, Atlantic and Mediterranean
sources are balanced (including backdoor cold fronts as Mediterranean) (Table 4). Distance to the Mediterranean and
elevation are important factors in determining the frequency of rainfall associated with backdoor cold fronts. Thus, backdoor
cold fronts are associated with 38.78% of Mallorca rain events and are still frequent situations at the two sites from the
Iberian Range (20.6% in Ortigosa de Cameros and 23.9% in Molinos). The frequency of convective precipitation is higher at
the three mountain sites (20.6% in Ortigosa de Cameros, 24.3% in Molinos and 23% in Borrastre), compared to those sites at
lower elevation (17% in Oviedo; 11.9% in El Pindal, 17% in Barcelona and 20.4% in Mallorca).

The Kruskal-Wallis test was applied to investigate if there were significant differences in the $\delta^{18}O_p$ values of the three
rainfall types analysed (Atlantic, backdoor frontal precipitation, and convective) in the studied stations at the daily scale.
Test values shown in Table 5 ($p$ values < 0.05) indicate the $\delta^{18}O_p$ values of at least two of the three rainfall types are
significantly different (this does not apply for Oviedo and Barcelona since the degrees of freedom are too low to yield a
significant result). Thus, this means that the type of rainfall (frontal versus convective) is an important factor controlling
$\delta^{18}O_p$ values in the studied transect at daily scale. This result is also evident where the three rainfall types are represented
according to their $\delta^{18}O_p$ composition (Figure 7). Thus, regarding $\delta^{18}O_p$ composition, convective precipitation (in green in
Figure 7) is associated with the highest $\delta^{18}O_p$ values, while events related to Atlantic and backdoor cold fronts display more
negative $\delta^{18}O_p$ values (albeit with a large spread), consistent with previous studies (Aggarwal et al., 2016). The highest $\delta^{18}O_p$
values associated with convective precipitation may relate to the critical role played by the re-evaporation of droplets, a
circumstance that usually takes place during convective rainfall (Bony et al., 2008). In any case, what is relevant here, is the
similarity among $\delta^{18}O_p$ values of the two types of frontal rains (Atlantic fronts and Mediterranean backdoor cold fronts)
while there is a difference considering the type of precipitation.


Besides $\delta^{18}O_p$ values associated with the three rainfall types, variations of air temperature and precipitation have an effect in separating the three rainfall types (Figure 7). Regarding air temperature, backdoor cold front events are the ones occurring with colder temperatures while convective rains are more associated with the warm season. Thus, air temperature (and its

variation along a vertical profile) is another variable clearly associated with the type of rainfall, with higher temperature during convective rains and lower for the Atlantic and backdoor types. This is a clear reflection of the seasonal pattern of convective rains, which are more abundant in summer months (Table S1) thus preventing an isolation of the effect of the type of rainfall which appears mixed with the temperature effect. In contrast, the high number of outliers in the box plots of the amount of precipitation when organized by rainfall type (Figure 7) indicates that this parameter is determined more by

local factors (e.g.. topography) than by the specific synoptic situation.

**5 Conclusion**

The major findings in this study are summarized as follows:

- The analysis of $\delta^{18}O_p$ and $\delta^2H_p$ at seven stations along a west-to-east transect in northern Iberia and Balearic Islands yields similar LMWLs but all with lower slope and intercept values than the GMWL.

- Oviedo/El Pindal and Mallorca/Barcelona rainfall samples display the least negative $\delta^{18}O_p$ and $\delta^2H_p$ values in the transect. Our results suggest that this similarity in the two opposite stations (the westernmost ones and the easternmost ones) are due to, in the first case, the initial condensate of water vapour generated over the North

Atlantic and, in the second case, the influence of air masses originating in the Mediterranean Sea together with much warmer temperatures than in the other sites. Besides those effects, the "elevation effect" must be taken into account to explain the more negative average values at the three mid-transect stations (Ortigosa de Cameros, Borrastre and Molinos).

- The seasonal variability is larger at Ortigosa de Cameros, Borrastre and Molinos while in Oviedo-El Pindal is

reduced due to the single origin of rainfall in that area.

- Air temperature appears to be the most significant influence on $\delta^{18}O_p$ at daily and monthly scales with the highest air temperature-$\delta^{18}O_p$ dependency found for the Pyrenean station (slope of 0.38‰/°C), while only few sites in the transect show a significant negative correlation (monthly in Molinos; daily in Ortigosa de Cameros, Molinos, Barcelona and Mallorca) with precipitation amount.

- The dominance of rainfall with an Atlantic origin is clear in the study of rainfall back trajectories associated with each event analysed. Additionally, the distance travelled inland in a quite dry region also conditions the recycling of re-evaporated moisture providing final enriched $\delta^{18}O_p$ values.

- Convective rainfall yields higher $\delta^{18}O_p$ values, while rainfall events related to Atlantic and backdoor fronts display more negative $\delta^{18}O_p$ values.



In conclusion, the northern Iberian region, is under the influence of two climatic regimes (Atlantic and Mediterranean) and affected by different moisture sources. Therefore, synoptic-scale atmospheric circulation is playing a key role in determining the ranges, values and seasonal distribution of $\delta^{18}O_p$ variability. Future detailed studies focusing on particular events that can be traced along the whole west-to east transect will be conducted to further understand the air masses trajectories over northern Spain and their influence on $\delta^{18}O_p$ variability.


**Data availability**

All data are included in the Supplementary Tables S1 and S2.

**Author contribution**

The paper was conceived by AM, CPM, MB, CS, HS and IC. MI carried out the back trajectory study and CAM provided the synoptic patterns during rainfall days. JF, CO, AM and ADH contributed to rainfall sampling and/or isotopic analyses. IB and FV helped with data interpretation. All authors contributed to the writing of the paper.

**Competing interests**

The authors declare that they have no conflict of interest

**Acknowledgements**

We acknowledge CTM2013-48639-C2-2-R (OPERA), CGL2016-77479-R (SPYRIT) and PID2019-106050RB-100 (PYCACHU) projects for main funding. Part of the previous isotopic analyses were carried out in the framework of GA-LC-030/2011, CGL2010-16376 and CGL2009-10455/BTE projects. This work is a contribution of Geomorfología y Cambio Global, Geotransfer and PaleoQ (IUCA) research groups (Aragón Government). We are extremely grateful to all people who carried out the rainfall sampling: Emilio (Molinos, Teruel); Ramiro Moreno (Borrastre, Huesca); M. Angeles, Sara and Juan
(Ortigosa de Cameros, La Rioja); Montse Guart (Barcelona); Alejandro Gallardo and Joan Fornós (Manacor, Mallorca). The Ebro Hydrographic Confederation network (SAIH www.saihebro.com) and the NOAA database are acknowledged for providing, respectively, meteorological data and software for the back trajectory analyses (HYSPLIT). We thank Georgina Mateu of the University of Barcelona for their temperature and precipitation data. We also acknowledge the www.meteo4u.com, www.meteoclimatic.net, http://www.larioja.org/emergencias-112/es/meteorologia and
http://balearsmeteo.com websites and the Sant Llorenç des Cardassar observatory for the meteorological data in this region





and the European Centre for Medium-Range Weather Forecasts for the ERA-Interim dataset. IC also thanks the ICREA Academia program from the Generalitat de Catalunya. We dedicate this study to our colleague Carlos Sancho who intensively worked to produce this large $\delta^{18}$O rainfall dataset for northern Iberia.

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

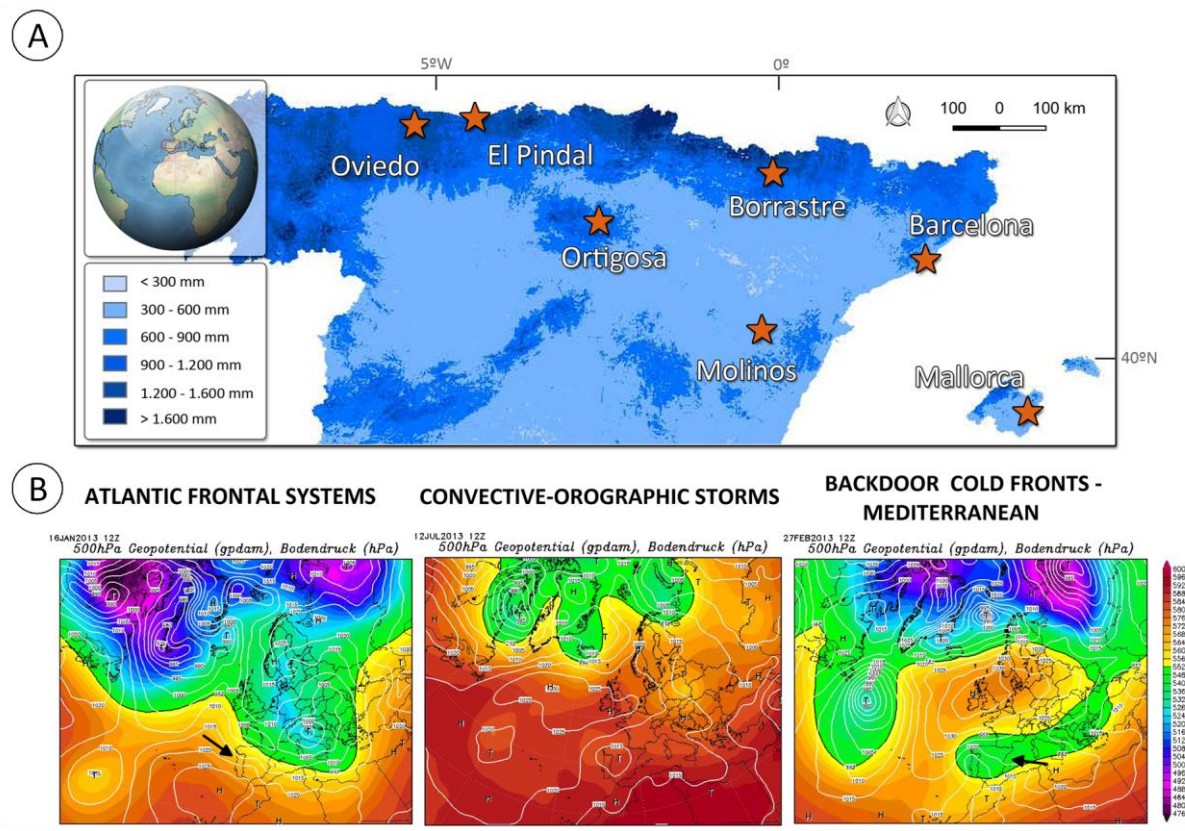

**Figure 1. (A) Location of the studied stations in northern Spain where rainfall was collected. Map source: Average annual precipitation (mm) for 1980-2005 provided by the Spanish Ministry of Agriculture and Fisheries, Food and Environment (MAPAMA); (B) weather maps showing the three precipitation regimes of the IP defined by Millán et al. (2005): (i) Atlantic frontal systems, (ii) convective–orographic storms, and (iii) Backdoor cold fronts from the Mediterranean Sea. In the maps, the sea level pressure and the 500 hPa geopotential height (gpdam in German) are indicated by the different colors; the scale represents the height- from 4600 to 6000 m - where the pressure of 500hPa is reached. White lines are the isobars (bodendruck in German). Source: CFS Reanalysis (CFSR) and Wetterzentrale.**





**Figure 2. Scatter plots of $\delta^2H_p$ versus $\delta^{18}O_p$ in precipitation and Local Meteoric Water Lines (LMWL), including equations, for El Pindal, Ortigosa de Cameros, Borrastre, Molinos and Mallorca with Barcelona stations. Note that El Pindal plot includes only 36 samples since $\delta^2H$ was not measured in the remaining ones. The difference in the other graphs in sample number (n) respect to those indicated in Table S1 is due to the removal of some samples that have been subject to evaporation effects (see text for more information). Global Meteoric Water Line (GMWL) and Western Mediterranean Meteoric Water Line (WMMWL) are plotted in black and gray, respectively, in every graph.**







**Figure 3. Event $\delta^{18}O_p$ time series for the studied stations presented versus time (2010-2017). Note that El Pindal samples are not represented since they do not overlap with the other stations (2006-2009). See text for more explanation.**

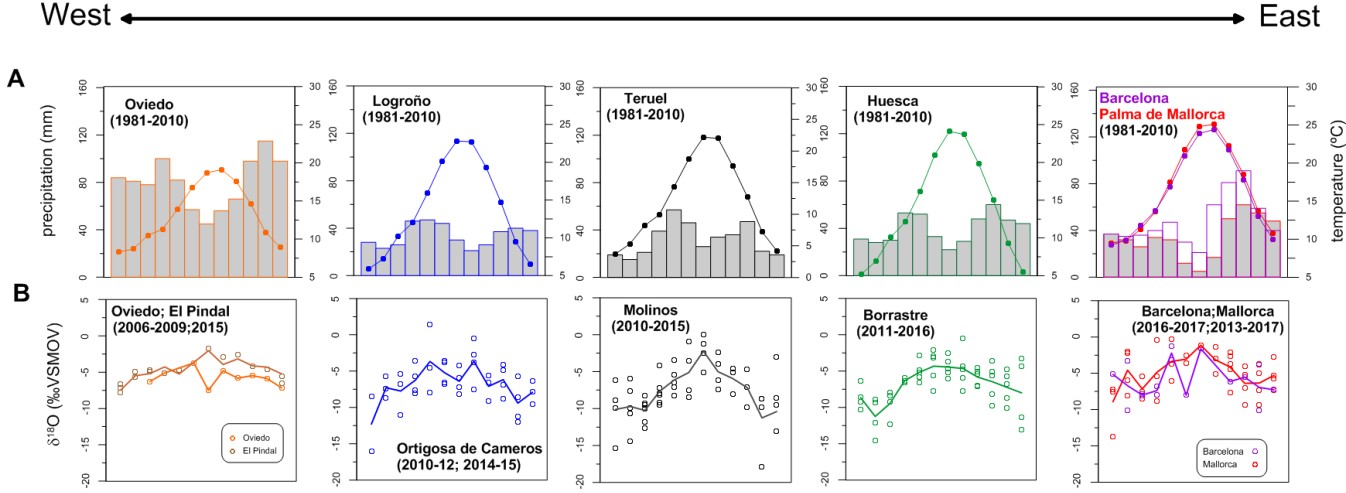

**Figure 4. (A) Climographs showing monthly mean temperature (line with dots) and monthly mean rainfall (bars) obtained for the longest AEMET meteorological stations available next to the study sites (Oviedo, Logroño, Teruel, Huesca, Barcelona and Palma de Mallorca). Note that these stations are not at the same elevation or microclimate as the ones where rainfall was collected. For this reason, the climographs are indicated here to account for broad regional climates while the correlations (Table3) with meteorological data were performed using more proximal (although shorter in the recorded time interval) stations. (B) Variability of monthly weighted $\delta^{18}O_p$ at the studied sites. Dots represent monthly precipitation-amount weighted averages and lines are the mean of these monthly precipitation-amount weighted averages (see also Table 2 and Table S2).**





**Figure 5. Wind roses showing the averaged back trajectories directions whose air masses produced precipitation in three stations in northern Iberia: Oviedo (northern Spain), Borrastre (central Pyrenees) and Mallorca (Balearic Islands). (A) Amount of precipitation (measured at the nearest meteorological station) during the intervals of sample collection and (B) $\delta^{18}O_p$ indicated by colors (see legends). Source regions of each air mass, generated by averaging the direction of each point of the back trajectory (20 points), are broken into 10º sectors. The percentages of back trajectories, whose averaged directions are associated with each 10º sector, are shown as dashed circles (from 0 to 12%).**





**Figure 6. Air mass history for selected days with precipitation at Borrastre site. The inserted maps indicate with more detail the trajectory over Iberia in every case. The three lines represent the air masses at different elevation (red: 850 hPa, blue: 700 hPa and green: 500hPa) (see text for more explanation).**




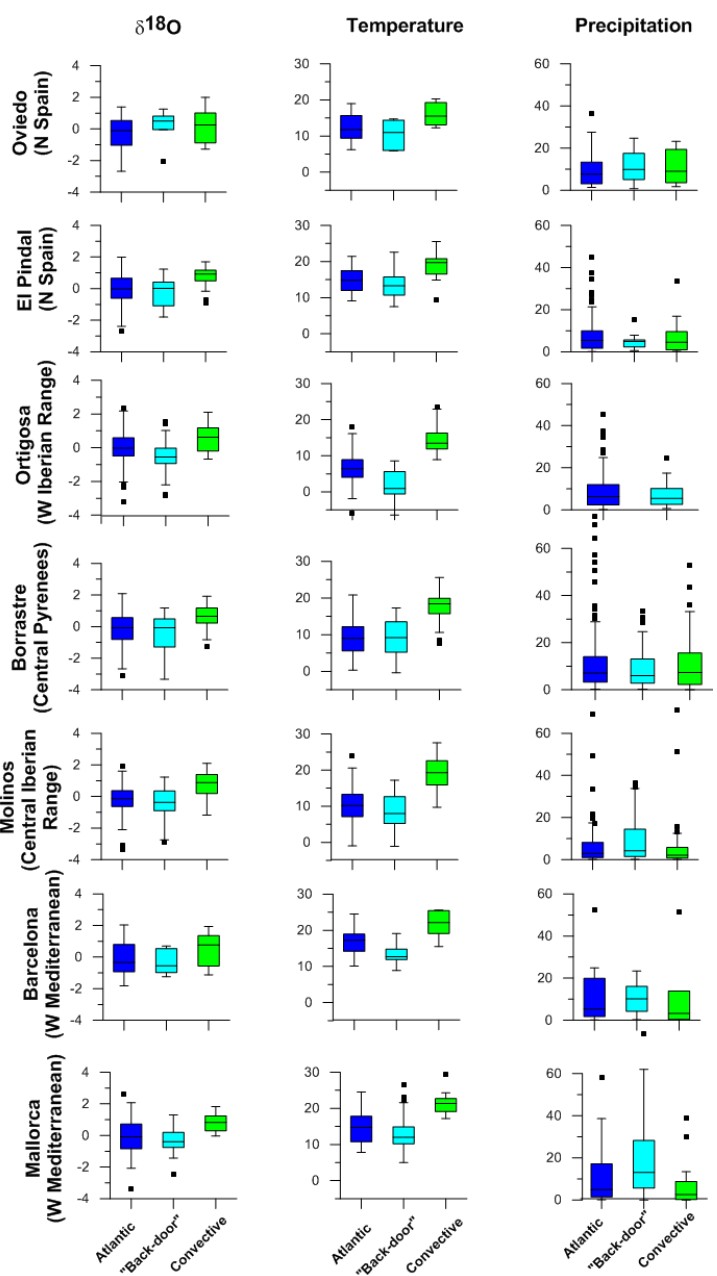

**Figure 7.** Box plots of $\delta^{18}O_p$, air temperature and precipitation amount for the three identified rainfall types in northern Iberia: Atlantic fronts (in dark blue), backdoor cold fronts (light blue) and convective precipitation (in green) for the studied stations. The central rectangle spans the first quartile to the third quartile (the *likely range of variation*, the *IQR*). A segment inside the rectangle shows the median and "whiskers" above and below the box show the locations of the minimum and maximum. The Kruskal-Wallis test indicates that at least two of the three rainfall types are significantly different in terms of their $\delta^{18}O_p$ values.



**Table captions**

**Table 1. Data collection details for the seven studied stations. KGC: Köppen and Geiger climate classification; AEMET: Agencia Española de Meteorología; SAIH: Automatic Hydrologic Information System. See Table S1 for all the isotopic and meteorological data. The AEMET stations with long series represented in Figure 4A are indicated.**

| | | Rainfall collection site | | | | | | | Meteorological data | | | | |
|---|---|---|---|---|---|---|---|---|---|---|---|---|---|
| *Location* | | *Coordinates and altitude* | | | *KGC* | *Data description* | | | *Station* | | *Annual mean Temp (°C)* | *Annual rainfall (mm)* | *AEMET long series (with data from 1981-2010)* |
| | | *Lat* | *Long* | *Altitude m asl* | | *Collection period* | *Nº samples* | *Laboratory* | *Name* | *Type* | | | |
| *Cantabrian coast* | *Oviedo* | 43°21N | 5°51W | 245 | Cfb | Feb 2015-Jan 2016 | 47 | Universities. of Oviedo and Barcelona (UB) | Oviedo (120 km from El Pindal) | AEMET | 13.3 | 960 | Oviedo |
| | *El Pindal* | 43°23N | 4°31W | 24 | Cfb | Nov 2006-Feb 2007 July 2007-May 2008 Jan 2009-April2009 | 101 | | Data from reanalysis (ECWMF ERA interim data) | | | | |
| *Iberian Range* | *Ortigosa de Cameros* | 42°10N | 2°42W | 1060 | Dsb | Sep 2010–Dec 2014 | 189 | IACT-CSIC and UB | Villoslada de Cameros (6.5 km) | La Rioja govern | 9.6 | 650 | Logroño |
| | *Molinos* | 40°47N | 0°26W | 1040 | Dsb | March 2010-May 2015 | 254 | IACT-CSIC | Gallipuén (7 km) | SAIH Ebro | 12 | 500 | Teruel |
| *Pyrenees* | *Borrastre* | 42°29N | 0°06W | 770 | Csb | Since April 2011 | 374 | | Borrastre (in situ) | Meteo-climatic | 13.5 | 900 | Huesca |
| *Mediterreanean* | *Barcelona* | 41°21N | 2°06E | 20 | Csa | Since Oct 2015 | 53 | UB | Barcelona-Zona Universitaria (in situ) | meteocat | 17.2 | 430 | Barcelona |
| | *Mallorca (Manacor and Porto Cristo)* | 39°33N | 3°12E | 90 | Csa | Since May 2013 | 98 | | Sant Llorenç (8 km) | AEMET | 18.8 | 590 | Palma de Mallorca |




**Table 2. Mean values of δ$^{18}$O$_p$ data for the seven sites in the study transect at a monthly and annual scale. Only months and years with all events collected are averaged. Note that the number of months or years averaged (n, in the table) are not the same for all the stations, neither the time period considered (check Table 1 for the sampling period in every station). For the complete monthly dataset with all the monthly values indicated, please refer to Table S2.**


| δ$^{18}$O$_p$ (‰) | *Cantabrian coast* | | | | *Iberian Range* | | | | *Pyrenees* | | *Mediterranean* | | | |
|---|---|---|---|---|---|---|---|---|---|---|---|---|---|---|
| | **Oviedo** | | **El Pindal** | | **Ortigosa de Cameros** | | **Molinos** | | **Borrastre** | | **Barcelona** | | **Mallorca** | |
| January | n=0 | | n=3 | -7.46 | n=2 | -12.29 | n=4 | -10.12 | n=4 | -8.64 | n=1 | -5.15 | n=4 | -8.91 |
| February | n=0 | | n=2 | -5.49 | n=3 | -7.28 | n=5 | -9.71 | n=4 | -11.25 | n=2 | -6.71 | n=4 | -4.57 |
| March | n=1 | -6.29 | n=2 | -5.19 | n=3 | -7.74 | n=6 | -10.25 | n=4 | -9.49 | n=2 | -8.00 | n=3 | -7.15 |
| April | n=1 | -5.12 | n=2 | -4.27 | n=4 | -6.25 | n=6 | -7.68 | n=4 | -6.35 | n=2 | -7.38 | n=3 | -4.86 |
| May | n=0 | | n=1 | -5.25 | n=3 | -3.66 | n=5 | -6.13 | n=6 | -5.19 | n=2 | -2.21 | n=3 | -3.36 |
| June | n=1 | -3.73 | n=0 | | n=4 | -5.21 | n=4 | -5.12 | n=5 | -4.32 | n=1 | -8.01 | n=2 | -3.06 |
| July | n=1 | -7.50 | n=1 | -2.04 | n=4 | -6.39 | n=4 | -2.22 | n=5 | -4.44 | n=1 | -1.64 | n=1 | -1.13 |
| August | n=1 | -4.80 | n=1 | -3.94 | n=4 | -3.64 | n=4 | -5.00 | n=5 | -4.65 | n=0 | | n=3 | -3.15 |
| September | n=1 | -5.83 | n=1 | -3.17 | n=3 | -7.09 | n=3 | -5.93 | n=5 | -5.83 | n=1 | -6.13 | n=5 | -4.14 |
| October | n=1 | -5.47 | n=1 | -4.12 | n=4 | -6.17 | n=3 | -7.18 | n=5 | -6.46 | n=1 | -5.53 | n=4 | -6.38 |
| November | n=1 | -5.87 | n=1 | -4.31 | n=4 | -9.40 | n=4 | -11.26 | n=5 | -7.24 | n=2 | -6.95 | n=4 | -6.34 |
| December | n=1 | -7.16 | n=2 | -5.23 | n=3 | -7.91 | n=3 | -10.41 | n=4 | -8.00 | n=1 | -7.27 | n=4 | -5.28 |
| Annual | n=0 | | n=0 | | n=1 | -7.09 | n=2 | -7.18 | n=3 | -6.37 | n=0 | | n=0 | |





**Table 3. Spearman´s rank correlation coefficients between $\delta^{18}O_p$ and air temperature and precipitation amount for every sampling station at daily scale (A) and monthly scale (B) using deseasonalised data (removing the seasonal component by subtracting their monthly averages). Significant correlations with *p value* < 0.05 after application of *Bonferroni test* are in red. Note that the relatively small size of Oviedo and Barcelona rain events likely precludes statistically significant correlations. (C) Multiple regression coefficient (r) and *p-value* for every site is included, indicating the coefficient and the standard error for the constant, the precipitation and the temperature variables. As an example, the equation for Molinos should be read as follows: $\delta^{18}O_p$=–**

**0.05(±0.019) *A* +0.40(±0.05) *T* + 0.43, with *A* as the amount of precipitation, *T* as air temperature and 0.43 as a constant value.**

| | | Oviedo | El Pindal | Ortigosa de Cameros | Molinos | Borrastre | Barcelona | Mallorca |
|---|---|---|---|---|---|---|---|---|
| (A) Daily correlations | | n = 39 | n = 109 | n=189 | n=248 | n=352 | n=53 | n=98 |
| $\delta^{18}O_p$ - temperature | $r_s$ | 0.23 | 0.34 | 0.25 | 0.41 | 0.31 | 0.24 | 0.35 |
| | p value | 0.328 | 0.0012 | 0.001 | 2.00E-11 | 1.17E-09 | 0.21 | 0.0013 |
| $\delta^{18}O_p$ - precipitation amount | $r_s$ | -0.22 | -0.06 | -0.32 | -0.19 | -0.11 | -0.35 | -0.28 |
| | p value | 0.368 | 1 | 1.05E-05 | 0.005 | 0.119 | 0.029 | 0.013 |
| (B) Monthly correlations | | n = 9 | n = 17 | n=41 | n=51 | n=49 | n=16 | n=40 |
| $\delta^{18}O_p$ - temperature | $r_s$ | 0.3 | 0.33 | 0.46 | 0.76 | 0.61 | 0.39 | 0.41 |
| | p value | 1 | 1 | 0.013 | 3.36E-10 | 1.44E-05 | 0.804 | 0.05 |
| $\delta^{18}O_p$ - precipitation amount | $r_s$ | 0.066 | -0.44 | -0.34 | -0.4 | -0.11 | -0.30 | -0.12 |
| | p value | 0.843 | 0.4 | 0.176 | 0.018 | 1 | 1 | 0.436 |
| (C) Multiple regression (with daily data) | r | 0.30 | 0.40 | 0.40 | 0.43 | 0.30 | 0.32 | 0.41 |
| | p value | 0.118 | 0.0001 | 3.36E-08 | 4.68E-13 | 8.13E-09 | 0.004 | 0.008 |
| Constant | Coeff | 0.14 | 0.32 | -1.6 | 0.43 | -2.83E-11 | -0.49 | 0.23 |
| | Std err | 0.43 | 0.24 | 0.22 | 0.18 | 0.16 | 0.38 | 0.26 |
| Precipitation | Coeff | -0.015 | -0.013 | -0.11 | -0.05 | -0.018 | -0.05 | -0.02 |
| | Std err | 0.05 | 0.04 | 0.02 | 0.019 | 0.014 | 0.02 | 0.017 |
| Temperature | Coeff | 0.21 | 0.25 | 0.25 | 0.40 | 0.40 | 0.37 | 0.31 |
| | Std err | 0.11 | 0.05 | 0.06 | 0.05 | 0.06 | 0.19 | 0.11 |



**Table 4. Relative frequency (in %) of the three rainfall types in every studied station.**

| | | Cantabrian coast | | Iberian Range | | Pyrenees | Mediterranean | |
|---|---|---|---|---|---|---|---|---|
| | | Oviedo | El Pindal | Ortigosa de Cameros | Molinos | Borrastre | Barcelona | Mallorca |
| *Atlantic fronts* | | 68.09 | 71.29 | 58.7 | 51.8 | 65.2 | 58.49 | 40.82 |
| *Backdoor cold fronts* | | 14.89 | 16.83 | 20.6 | 23.9 | 11.8 | 24.53 | 38.78 |
| *Convective* | | 17.02 | 11.88 | 20.6 | 24.3 | 23.0 | 16.98 | 20.41 |

**Tabla 5. Kruskal-Wallis test performed on $\delta^{18}O_p$ data to discriminate if the three synoptic patterns are statistically different in**
**terms of their isotopic composition. High values of the test (Kruskal-Wallis H) and low *p-value*s indicate that at least two of the**
**three synoptic patterns are statistically different in terms of $\delta^{18}O_p$ data.**

| | Cantabrian coast | | Iberian Range | | Pyrenees | Mediterranean | |
|---|---|---|---|---|---|---|---|
| | Oviedo | El Pindal | Ortigosa de Cameros | Molinos | Borrastre | Barcelona | Mallorca |
| Kruskal-Wallis *H* | 3.017 | 10.86 | 23.3 | 48.38 | 47.84 | 4.109 | 22.23 |
| *p* value | 0.221 | 0.004 | 8.7E-06 | 3.12E-11 | 4.09E-11 | 0.1282 | 1.49E-05 |






**Supplementary**


**Figure S1. Wind roses represent the averaged back trajectories of air masses that produced precipitation at three stations in northern Iberia: Oviedo (northern Spain), Borrastre (central Pyrenees) and Mallorca (Balearic Islands). Trajectories shown were computed for only 24 hours.**

As supplementary

**Table S1. Event $\delta^{18}O_p$ and $\delta^2H_p$ data for the stations considered in this study. Meteorological data from nearby stations (Table 1) are also included.**


As supplementary

**Table S2. Monthly $\delta^{18}O_p$ data for the stations considered in this study**

As supplementary