# Peer review of "Measurement reports"

_Atmospheric Chemistry and Physics, 2020_

## Referee Comment (RC2)

**Review of "Spatial variability of northern Iberian rainfall stable isotope values: Investigating climatic controls on daily and monthly timescales"**
**by Moreno et al.**

This paper presents a set of multi-year time series between 2010 and 2017 of daily isotope data in precipitation from six station in the northern Iberian Peninsula. This high-quality dataset is very valuable for the isotope community, in particular because it is compiled at high temporal (daily) resolution and comprises multiple years with simultaneous data from several stations. In addition to the very valuable dataset, which could also be submitted to a dedicated data journal such as ESSD, this paper discusses several relevant meteorological factors for the large day-to-day variability in the collected isotope data. As the authors write in their introduction, an in-depth understanding of the regional climatic controls on modern precipitation water isotopes is still missing, but is of key importance for a better understanding of terrestrial climate proxies. The interesting and important discussion on the climatic controls of the isotope composition of precipitation is what makes this paper fit in principle into the scope of ACP. However, a substantial effort is needed to address several important structural and methodological weaknesses in this paper. I consider this paper acceptable for publication if the following major comments can be addressed adequately:

1) **Shorten and strengthen the structure of the discussion on the climatic controls**: This paper is rather long and there is frequent repetition of facts that are in my opinion of (very) minor importance such as the difference in oceanic surface water isotope composition that is mentioned at lines 292ff, 309ff, 369ff, 387ff, and 442ff. As the authors mention themselves, I believe the effect of such small differences (order of 0.5‰) in $\delta^{18}O$ has a negligible impact on precipitation isotopes. Much rather, I think, the authors should mention differences in moisture source conditions (temperature, relative humidity with respect to sea surface temperature) at the location of the moisture uptake as an important factor. This has been shown by many recent studies that focus on water vapour in the marine boundary layer and/or on downstream precipitation (e.g. Dansgaard, 1964; Craig and Gordon 1965; Pfahl and Wernli 2008; Steen-Larsen et al. 2015; Aemisegger, 2018; Thurnherr et al. 2020).
My concrete suggestion to remediate this is:
   a) To mention the aspect of the difference in ocean surface isotope composition once, but only shortly, in section 5.3 and to add a discussion about the other, more important moisture source controls, which I detailed above.
   b) To strengthen the structure of the paper by not mentioning the temperature and precipitation amount effect at multiple locations (e.g. in Section 5.2 and p. 16 l 487-495), but to confine this analysis to Section 5.2.
2) **Temperature and precipitation amount controls on the variability of the isotopic composition of precipitation**: (e.g., P. 2, L. 34; Section 5.2, P.16 L487-495) I am a bit unhappy with these statements about the "air temperature exerting the most significant influence on $\delta^{18}O_p$". Based on a correlation analysis it is not possible to infer any mechanistic control or directed influence. In my opinion this "temperature effect" is very complex and not yet so well understood. Usually, it is conceptualised by using a simple air parcel model with continuous uptake in the marine boundary layer and progressive rain out thereafter following a Rayleigh distillation model approach. To formulate it a bit provocatively: it could also be the $\delta^{18}O$ (or much

rather the amount of remaining water vapour, i.e. the specific humidity) that is actually controlling the temperature. To me, such correlations just indicate an overarching coherence in the hydroclimate system, in which a change in one variable necessarily implies concurrent changes in many of the others. Progressive rain out along an air parcel is one very good example, in which this seems to be the case to me. I would encourage the authors to formulate the result of their correlation analysis more cautiously, with this thinking in mind and to avoid inferring any influence without precise description of the implied process.

3) **Moisture source identification discussion:** there is substantial literature that investigates the moisture sources of precipitation and water vapour with sophisticated methods based on Lagrangian moisture source diagnostics (e.g. Sodemann et al. 2008, Gimeno et al. 2010) or Eulerian tracer studies (e.g. Winschall et al. 2014). The method adopted here, with only three trajectories calculated per station accompanied by the wind rose approach to quantify source regions is not adequate for addressing the question of the moisture source of precipitation at the different stations. What I suggest here are the following changes:
   a. Reword all places where moisture sources are mentioned and formulate it as the "origin of the air parcels associated with precipitation" or a similar wording.
   b. Recompute trajectories (vertically stacked from the surface to 300 hPa), and only select those that are associated with a relative humidity of >80% at the arrival (station) and then compute the wind roses. For the wind roses, the relevant timescale for the largest part of the moisture uptakes (in particular in summer) is up to 5 days before arrival (e.g. Läderach and Sodemann, 2016). Doing the wind roses separately for 0 to 5 days before arrival (in the paper) and 5 to 10 days before arrival (in the supplement) would seem adequate. This will most probably provide a better rough estimate of the precipitating air parcels' moisture uptake region.

4) **Seasonality of moisture sources:** Many climatological moisture source studies have highlighted a strong contrast in land vs. ocean sources for continental precipitation in winter vs. summer. A strong seasonal cycle is usually observed with a dominant contribution of continental moisture recycling in summer and a much more important contribution of oceanic sources in winter. A recent study analysing the trace element composition of precipitation in the Pyrenees (Suess et al., 2020) also shows this tendency. This seasonality in land vs ocean moisture source contribution to Iberian precipitation is most certainly an important driver of the seasonal cycle of the isotope signals in precipitation. This key aspect should be mentioned upfront in the introduction around lines 72-73.

5) **Meteorological context of precipitation events:**
I find the discussion around the meteorological context of precipitation events very important and interesting. However, it would be great if the authors could write a bit more precisely, how they expect the different precipitation mechanisms (frontal systems in winter, convective precipitation in high pressure systems in summer) to impact the isotope signature of precipitation. There are a few studies that analysed isotope signals during cold frontal passages in Europe (Aemisegger et al. 2015, Graf et al. 2019) and squall lines in Africa (Risi et al. 2008), as well as a climatological study over Europe (Christner et al. 2018) addressing the impact of continental moisture

recycling on isotope signals in vapour and precipitation. Furthermore, the impact of convection during a cold front passage in the Mediterranean was investigated in a modelling study by Lee et al. 2019. The recent paper by Rüdisühli et al. 2020 would provide the authors some guidelines, about which weather system dominates precipitation in which season on the Iberian Peninsula (cold fronts in winter, high pressure systems in summer).

Minor comments:

1) P. 1, L. 29: **the** rainfall isotopic variability
2) P. 1, L. 30: determining **the** rainfall isotopic variability
3) P. 2, L. 32: "Atlantic fronts are found to be the dominant source": Fronts delimit two different air masses with contrasting thermodynamic properties (e.g. warm and moist air mass from a cold and dry air mass). They are not "sources" of rain events. I suggest to rewrite this into something like: "Frontal systems associated with North Atlantic cyclones are the dominant mechanism inducing precipitation over Northern Iberia, in particular in winter".
4) P. 2, L. 37: why "but the type of precipitation…" shouldn't it be "in addition".
5) P. 2: It is a bit unclear in the abstract which factors the authors think are the most important. Aren't the investigated aspects overlapping to some extent and just show consistent relations but from different angles? To me the continentality aspect is very much related to the source aspect and the type of mechanism inducing rainfall is related to the precipitation amount and the temperature. I would recommend the authors to revise the abstract and structure it in a clearer way.
6) P. 2, L. 47: in modulating $\delta^{18}O_p$ **in** a particular region
7) P. 3, L 67: for example (Aggarwal et al., 2016). Regrettably
8) P.4, L109- 122: While I would find it very interesting to investigate the role of cyclones and the interannual variability of the North Atlantic storm track on the isotope signals on the Iberian Peninsula, I think this discussion about the NAO is a bit out of scope here. It is not taken up later on in the analyses performed.
9) Section 3.1: Given the importance of the sampling procedure, it would be nice if the authors could roughly quantify the uncertainty associated with their sampling system. A recent study, that has performed a targeted uncertainty analysis in this respect is Fischer et al. 2019.
10) P. 7. L. 200: **no** good data
11) P. L 225ff: I did not understand how the authors proceeded here, only when reading their results (P. 13, L. 392-399). Lines 392-399 are explaining the method and should be brought forward to the methods section and be removed from the results.
12) P. 8, L. 232: I did not understand, how the authors used the disaggregated precipitation time series and why this was important.
13) Figure 3 should be split into different panels with one panel per station. Currently it is very difficult to distinguish the different time series. Furthermore, I would urge the authors to add the time series of $\delta D$, the deuterium excess and precipitation totals. They are equally important.
14) P. 8, L 253: Here I think the authors could make it clearer that overall, there is a very large day-to-day variability in their isotope data that is as large as the seasonal cycle. This would emphasise the value of the high temporal resolution of their measurements.

15) P. 8, L. 255: over which time period were these sums accumulated?

16) P. 9, L. 264: "bi-model pattern" mention with peaks in Spring and autumn

17) P. 9, L. 265: remove "quite"

18) Section 5: in the short introductory paragraph of this section, it is important to add a statement about the fact that the hydrological cycle is complex and that many processes play a role in the formation of the isotope signals in precipitation ranging from source processes, transport processes, as well as cloud and rainfall formation at the sampling site. Furthermore, the authors should mention that they discuss several factors, that they believe play an important role, but they should also make it clear that these factors can play an overlapping role (i.e. due to the concurrent changes in many variables taking place along an air parcel's transport pathway, see my major comment 2).

19) P. 9, L. 281: why are moisture sources and air masses history mentioned here? They should be taken out of this sentence and it should be made clear in a subsequent remark, that the latitudinal location and the regional orography influences the circulation and therefore the air mass history.

20) P. 9, L. 287: it should be made clear here, which are the high elevation and which the low elevation sites.

21) P. 9, L. 289ff: this part should go to the source effect Section in 5.3.

22) P. 10, L. 298: "when **moist air** and clouds"

23) P. 10, L. 303: not the **main** synoptic pattern, see climatological analysis of the weather systems that induced precipitation in different seasons by Rüdisühli et al. 2020.

24) P. 10, L. 306-3012: this does not fit into this section

25) P. 10, L. 320: "separated from the Mediterranean"

26) P. 10, L 324: no comma before controlling factors

27) P. 11, L. 348: remove vertical, this is clear for deep convection

28) P. 11, L. 349: in the extratropics in summer

29) P. 11, L. 355: where a significant ... correlation is found.

30) P. 11, L. 356: I would not necessarily expect such a correlation at the monthly timescale

31) P. 11, L. 357: at the other sites

32) P. 12, L. 360: origin of air masses producing rainfall

33) P. 12, L. 361: also spatially variable in northern Iberia, these properties, and their relation to the observed $\delta^{18}O_p$ variability…

34) P. 12, L. 381: "with the Villars station"

35) P. 12, L. 381: "precipitation type and geographic origin" this is the topic of the next section

36) P. 12, L. 383: there are certainly much less precipitation events that are associated with fronts in summer than in winter

37) P. 12, L. 386: Remove the sentence about the isotope composition of the ocean at the moisture source and mention it only once (you do it already at lines 369ff).

38) P. 13, L. 389: Replace "moisture source" by "airmass origin"

39) P. 13, L. 409: I don't understand the sentence "In that way, it is clear the dominant WNW…".

40) P. 13, L. 415: what does more stable mean?

41) P. 13, L. 415-420: maybe a classification into subregions would make the discussion in the whole paper easier to follow.

42) P.13, L. 420: I of course agree that most probably the moisture source plays an important role, but the authors don't show this in a quantitative and methodologically convincing way.

43) P. 14, L. 435: "evaporated"-> evapotranspired

44) P. 14, L. 442: remove the reference to the composition of the ocean surface

45) P. 15: at several instances, the percentages should be indicated in integer precision and not floating numbers

46) P. 15, L. 462: when they can cause heavy precipitation and flooding

47) P. 16: I find the conclusions a bit disappointing in the sense that it is simply a list of findings and there is no opening of the study to new questions that have raised from this study. The authors should rewrite their conclusions in the light of their revisions of the structure and line of interpretation of the climate signals reflected by their isotope data.

48) P. 16: first conclusion: what does this mean for dexcess? I would be much more interested in seeing the precipitation deuterium excess signals and a bit more discussion on them (dexcess is a frequently used proxy for the conditions at the moisture source) instead of the many repetitions of the role of the ocean surface isotope composition at the moisture source.

**References:**

Aemisegger, F., On the link between the North Atlantic storm track and precipitation deuterium excess in Reykjavik, Atmos. Sci. Lett., 19:e865, doi:10.1002/asl.865, 2018.

Aemisegger, F., Spiegel, J. K., Pfahl, S., Sodemann, H., Eugster, W., and Wernli, H., Isotope meteorology of cold front passages: A case study combining observations and modeling, Geophys. Res. Lett., 42, 5652– 5660, doi:10.1002/2015GL063988, 2015.

Christner, E., Aemisegger, F., Pfahl, S., Werner, M., Cauquoin, A., Schneider, M., Hase F., Barthlott S., and Schädler G., The climatological impacts of continental surface evaporation, rainout, and subcloud processes on $\delta$D of water vapor and precipitation in Europe. Journal of Geophysical Research: Atmospheres, 123, 4390–4409, doi:10.1002/2017JD027260, 2018.

Craig, H. and Gordon, L., Deuterium and oxygen 18 variations in the ocean and the marine atmosphere, in: Proceedings of the Stable Isotopes in Oceanographic Studies and Paleotemperatures, 1965.

Dansgaard, W., Stable isotopes in precipitation, Tellus, 16, 436–468, doi:10.1111/j.2153-3490.1964.tb00181.x, 1964.

Fischer, B. M. C., Aemisegger, F., Graf, P., Sodemann, H., and Seibert, J., Assessing the sampling precision of a low-tech low-budget volume-based rainfall sampler for stable isotope analysis, Front. Earth Sci., 7:244, doi:10.3389/feart.2019.00244, 2019.

Gimeno, L., R. Nieto, R. M. Trigo, S. M. Vicente-Serrano, and J. I. López-Moreno, Where Does the Iberian Peninsula Moisture Come From? An Answer Based on a Lagrangian Approach. *J. Hydrometeor.*, **11**, 421–436, https://doi.org/10.1175/2009JHM1182.1, 2010.

Graf, P., Wernli, H., Pfahl, S., and Sodemann, H., A new interpretative framework for below-cloud effects on stable water isotopes in vapour and rain, Atmos. Chem. Phys., 19, 747–765, https://doi.org/10.5194/acp-19-747-2019, 2019.

Läderach, A., and  Sodemann, H.,  A revised picture of the atmospheric moisture residence time, Geophys. Res. Lett.,  43,  924– 933, doi:10.1002/2015GL067449, 2016.

Lee, K.-O., Aemisegger, F., Pfahl, S., Flamant, C., Lacour, J.-L., and Chaboureau, J.-P., Contrasting stable water isotope signals from convective and large-scale precipitation phases of a heavy precipitation event in Southern Italy during HyMeX IOP 13, *Atmos. Chem. Phys.*, 19, 7487-7506, doi:10.5194/acp-19-7487-2019, 2019.

Pfahl, S., and  Wernli, H.,  Air parcel trajectory analysis of stable isotopes in water vapor in the eastern Mediterranean, J. Geophys. Res.,  113, D20104, doi:10.1029/2008JD009839, 2008.

Rüdisühli, S., Sprenger, M., Leutwyler, D., Schär, C., and Wernli, H.: Attribution of precipitation to cyclones and fronts over Europe in a kilometer-scale regional climate simulation, Weather Clim. Dynam., 1, 675–699, https://doi.org/10.5194/wcd-1-675-2020, 2020.

Sodemann, H.,  Schwierz, C., and  Wernli, H.,  Interannual variability of Greenland winter precipitation sources: Lagrangian moisture diagnostic and North Atlantic Oscillation influence, J. Geophys. Res.,  113, D03107, doi:10.1029/2007JD008503, 2008.

Steen-Larsen, H. C.,  Sveinbjörnsdottir, A. E.,  Jonsson, Th.,  Ritter, F.,  Bonne, J.-L.,  Masson-Delmotte, V., Sodemann, H.,  Blunier, T.,  Dahl-Jensen, D., and  Vinther, B. M.,  Moisture sources and synoptic to seasonal variability of North Atlantic water vapor isotopic composition. J. Geophys. Res. Atmos.,  120, 5757– 5774. doi: 10.1002/2015JD023234, 2015.

Suess, E., Aemisegger, F., Sonke, J., Sprenger, M., Wernli, H., and Winkel, L., Marine versus continental sources of iodine and selenium in rainfall at two European high-altitude locations, *Environ. Sci. and Technol.*, 19;53(4):1905-1917, doi: 10.1021/acs.est.8b05533, 2019.

Thurnherr, I., Kozachek, A., Graf, P., Weng, Y., Bolshiyanov, D., Landwehr, S., Pfahl, S., Schmale, J., Sodemann, H., Steen-Larsen, H. C., Toffoli, A., Wernli, H., and Aemisegger, F., Meridional and vertical variations of the water vapour isotopic composition in the marine boundary layer over the Atlantic and Southern Ocean, Atmos. Chem. Phys., 20, 5811–5835, https://doi.org/10.5194/acp-20-5811-2020, 2020.

Winschall, A., Sodemann, H., Pfahl, S., and Wernli, H.,  How important is intensified evaporation for Mediterranean precipitation extremes?, J. Geophys. Res. Atmos.,  119,  5240- 5256, doi:10.1002/2013JD021175, 2014.

---

## Referee Comment (RC1) · Anonymous Referee #1 · 2 Nov 2020

General comments

This manuscript seeks to determine key factors responsible for daily and monthly rainfall d18O in northern Iberia. Using new multi-year record of the water isotopes in daily based precipitation at 7 stations in northern Iberia, they identify the key factors controlling the spatiotemporal d18O variability through a comparison with geographical and meteorological items and an analysis of air mass sources and precipitation types. Then they concluded that the relative contribution of each air mass (Atlantic, Mediterranean and continental) is a dominant factor of d18O variability in Conclusion, but in Abstract, they highlighted the role of precipitation type as a key factor (I repeatedly read this draft,

but I can't understand which factor they want to emphasize). Also, they noted that the other factors such as distance from the coast, altitude, temperature and precipitation amount also significantly contribute on d18O variability.

This paper presents a valuable data set of daily basis precipitation isotopes observed over northern Iberia. The new observation to cover the sparse region is valuable for isotope community. However, the purpose of this study does not seem to fit the scope of the ACP; The journal scope is focused on studies with general implications for atmospheric science rather than investigations that are primarily of local or technical interest. According to Introduction, the application of this study is to improve our interpretation of d18O proxy record. In addition, the research target (northern Iberia) is very specific. So, I'm wondering if ACP is the appropriate journal for this study. Frankly speaking, this manuscript is not the scientific manuscript, but the data report. The main task of this draft is just comparison with various factors such as temperature, precipitation amount, air mass trajectories, precipitation types etc. And most of the discussion is speculation without solid facts. For example, in section 5.3 the authors discussed the influence of moisture source effect, but their conclusion is based only on a single event (see Fig 6) and no explanation to show that their selected each event is typical (L427-L445). For these reasons, I unfortunately cannot recommend this manuscript for publication.

---

## Referee Comment (RC3) · Anonymous Referee #3 · 17 Feb 2021

This study investigates the isotopic variability in rainfall in the northern Iberian Peninsula at daily and monthly time scales. Several climatic and geographic factors that can help to explain isotope composition are discussed, such as temperature, amount of precipitation, the site elevation, and continental effects. Also, the study aims to links the isotopic variability to air mass pathways (i.e., moisture source effect) using back trajectories and rainfall types (frontal versus convective) based on different weather regimes of precipitation.

Overall, I found the study very interesting and potentially relevant. The study combines isotope measurements in rainfall with a meteorological perspective that can help to

explain the isotope variability along a transect of the northern Iberian Peninsula where different weather systems and processes can have an alternating influence on precipitation formation, potentially supporting an improved interpretation of paleoclimate archives in the region. However, I also have a few major concerns that need to be addressed before considering potential publication. In its current state, the storyline of the manuscript is at several places hard to follow, with discussion of results placed in different (sub)sections. In addition, the quality of the text allows for improvements. Please, see for more details general comment #1 and several other specific and technical comments. Another main concern is related to the analysis of the back trajectories that aim to provide an indication of the moisture source / air mass origin and the linkage of this origin to the isotope signals in precipitation. If my understanding is correct, the idea is that moisture pathways from the Atlantic leave a different isotopic imprint than those arriving from the east over the Mediterranean Sea. This analysis, however, seems somehow problematic as for all sites in the study area by far the larger part of trajectories arrive from the west / northwest. In addition, the manuscript states at several occasions that the isotope signals on the western and eastern margins of the Iberian Peninsula are relatively similar. As a consequence, the presented results in section 4.3 appear to be not in line with the stated conclusions derived from this analysis. Please, see for more details general comment #2. My recommendation is to satisfactorily address these two major comments as well as the specific and technical comments, and to resubmitted a revised manuscript for consideration of publication.

Please also note the supplement to this comment:
https://acp.copernicus.org/preprints/acp-2020-861/acp-2020-861-RC3-supplement.pdf
* * *
[Figure]

**Supplement:**

**Review of acp-2020-861**

Title: "Spatial variability of northern Iberian rainfall stable isotope values: Investigating climatic controls on daily and monthly timescales"

Authors: Moreno et al., 2020.

**Summary**

This study investigates the isotopic variability in rainfall in the northern Iberian Peninsula at daily and monthly time scales. Several climatic and geographic factors that can help to explain isotope composition are discussed, such as temperature, amount of precipitation, the site elevation, and continental effects. Also, the study aims to links the isotopic variability to air mass pathways (i.e., moisture source effect) using back trajectories and rainfall types (frontal versus convective) based on different weather regimes of precipitation.

Overall, I found the study very interesting and potentially relevant. The study combines isotope measurements in rainfall with a meteorological perspective that can help to explain the isotope variability along a transect of the northern Iberian Peninsula where different weather systems and processes can have an alternating influence on precipitation formation, potentially supporting an improved interpretation of paleoclimate archives in the region. However, I also have a few major concerns that need to be addressed before considering potential publication. In its current state, the storyline of the manuscript is at several places hard to follow, with discussion of results placed in different (sub)sections. In addition, the quality of the text allows for improvements. Please, see for more details general comment #1 and several other specific and technical comments. Another main concern is related to the analysis of the back trajectories that aim to provide an indication of the moisture source / air mass origin and the linkage of this origin to the isotope signals in precipitation. If my understanding is correct, the idea is that moisture pathways from the Atlantic leave a different isotopic imprint than those arriving from the east over the Mediterranean Sea. This analysis, however, seems somehow problematic as for all sites in the study area by far the larger part of trajectories arrive from the west / northwest. In addition, the manuscript states at several occasions that the isotope signals on the western and eastern margins of the Iberian Peninsula are relatively similar. As a consequence, the presented results in section 4.3 appear to be not in line with the stated conclusions derived from this analysis. Please, see for more details general comment #2. My recommendation is to satisfactorily address these two major comments as well as the specific and technical comments, and to resubmitted a revised manuscript for consideration of publication.

**General comments**

**1. Organization of the manuscript and writing**
One of the major revisions the article may need is related to the flow and organization of the text. Chapter 4 describes in two sub sections the daily and monthly rainfall isotopic variability, respectively, without that much explanation or interpretation is given. Then, in section 5, called a discussion, while in fact continuing to present results, comes back to subjects that were initially described. As a reader, I felt lost and had difficulty to follow the storyline here. I recommend to reorganize the structure of chapters 4 and 5 into one chapter that presents the results, for example, 4.1 "daily rainfall isotopic variability" (presenting the local water lines), 4.2 "monthly rainfall isotopic variability" (discussing seasonality), and then 4.3 "the geographical controls on rainfall isotopic variability", etc., until section 4.6 "the influence of rainfall types on isotopes". Also, I recommend to discuss the local meteoric water line, currently at two places in the manuscript (lines 244-247 and 281-285) only once in one place, and the same for the analysis on the seasonality, please, see also the specific comments below.

In addition, the quality of the writing may be improved. For example, the excessive use of text in brackets should be avoided when possible. Several examples and other suggestions for improvements of the text are noted under the technical comments.

**2. Source regions and backward trajectories**

Although the analysis that addresses the air mass origin associated with the rain events based on the trajectory calculations is very interesting, and is presented in a very interesting and visually illustrative way (Figure 5), it seems that the method is not adequately able to represent the processes the study is aiming for. As the manuscript articulates in lines 399-409, westerly trajectories are dominant, and possibly air mass pathways can undergo moisture uptake in the short time period before arriving over the target areas. However, as the text acknowledges, this process may be well under-represented in this methodology. Therefore, it appears to me that the conclusions, as stated in the abstract "moisture source region (Atlantic versus Mediterranean) also significantly modulate the $d^{18}O_p$ values" (lines 36-37), the results "These results highlight the importance of moisture source ... ... transect of this study" (lines 420-421), and the conclusions "affected by different moisture sources" (line 521), appear not in line with the presented analysis that shows "… very few episodes associated with other directions (Figure 5). This low, almost negligible, presence of trajectories associated with Mediterranean air mass advections …", (lines 399-401). This is once more underlined by the analysis in section 4.4 that shows that Atlantic fronts and backdoor cold fronts have very similar isotopic signals, in contrast to the convective-orographic precipitation. For this reason, I have the impression that the stated conclusions are not in line with the presented results, and that the adopted methodology may not suitable for its purpose. One thought that came up, and may potentially alleviate the limitation in the methodology, is to limit the time of the backward trajectories to one or two days. This analysis appeared actually to be done (Fig S1), and seems more appropriate and togive more meaningful results, especially for the Borrastre site showing predominantly trajectories from southerly directions that are associated with more enriched precipitation. I would recommend to replace Figure 5 by Figure S1 and limit the analysis to one- or two-day backward trajectories. However, a key point of the study, the easterly advection of moisture from the Mediterranean is not substantially increasing and not showing up in the results. Have the authors considered to use only trajectories from 850 hPa and excluding those from 700 hPa? At which time instances of the rainfall days were the trajectories started? At present, I do unfortunately not have any other suggestions that may lead to a suitable methodology that may serve the purpose of this aspect

**Specific comments**

Line 2. The title refers to "climate controls" on the variability of isotopic composition in rainfall. The study itself though seems more to be focused on meteorological processes such as moisture pathways and weather regimes/precipitation types of rain days. Perhaps, the authors may consider to use or add another term such as "weather", "meteorological", or "atmospheric"?

Line 39-40. Perhaps, besides referring to the dataset, this concluding sentence may also refer to the analysis that helps to understand rainfall isotope variability in relation to meteorological / atmospheric processes and geographic influences?

Lines 73 and 74. The term "trajectories" is perhaps quite technical for the introduction. Instead, a term that refers to actual physical processes, such as "air mass origins" or "air mass transport" may be more appropriate.

Lines 80-85. This is a crucial paragraph as it outlines what the intention of the study is, and what it adds to previous studies as outlined in the text above. The thought behind the first sentence "In this paper …" is not clear to me. Is the approach, based on multiple stations new and is that the main selling point of the paper? Or is this study presenting a comprehensive analysis based on multiple stations across the Atlantic-Mediterranean transect? In the first case, the authors may write "we introduce a new approach…", and in the latter case, "we present a comprehensive / multiple perspective analysis on daily and monthly …". Also, is it really new that a study considers multiple stations across a region? If other studies followed such an approach, perhaps for other regions, this may deserve attention in the introduction to provide context for this study, for example by adding a new paragraph. In addition, this paragraph may explicitly refer to the atmospheric processes and geographic factors that influence the isotopic rainfall variability that are addressed in this study to guide the reader's expectations.

Line 87. This section addresses besides the site description and climate also the different weather regimes that bring precipitation over the northern Iberian Peninsula. This may be reflected in the title of the section.

Lines 103-104. The phrase "also easterly advections over the Mediterrean Sea" sounds somewhat vague. Please, rewrite, perhaps in the direction of "fronts that approach the Iberian Peninsula from the east (backdoor cold fronts)"..

Lines 119-122. While reading this paragraph I somehow lost the storyline. The first sentence refers to the dominant source regions and seems to follow as a conclusion from the text above, while the next sentence introduces the four different climate zones. The authors may consider to add the first sentence to the paragraph above (or elsewhere), and to start a new paragraph with the second sentence. Then, the introduction of the four climate zone regions is hard to follow; It may help rephrase this sentence as, for example, "Below, the seven stations are grouped into four regions and described in terms of their climatology". Also, it feels somewhat chaotic to refer at this stage multiple times to Figure 4 while Figures 2 and 3 have not yet been discussed. Is it necessary to include the line "Regional meteorological data are provided in Figure 4A."?

Lines 123-127. Can this paragraph be shortened by saying "The sites of El Pindal and Oviedo…" and removing the sentence on lines 126-127 "Additionally, … in this study."?

Line 197. To what "Meteorological data" is referred? If this is the air temperature and precipitation, please, remove the brackets, and rephrase the sentence to place more emphasis on these meteorological variables, for example, as "Air temperature and precipitation are obtained from the closest meteorological stations over the sampling periods, as indicated in Table 1, to investigate …. ".

Line 292. Usually, when referring to the ERA-Interim analysis Dee et al. (2011) is cited.

Lines 211-238. In this paragraph I feel quite overwhelmed by the many references to Tables and Figures for which here only the applied methodology is described (e.g., Tables 3, 4, and 5 and Figure 5). I would recommend to only refer explicitly to the Tables and Figures when discussing the scientific results, not when describing the used (statistical) methods.

Lines 223-224. Which reanalysis data are the HYSPLIT simulations using? This should briefly be mentioned, including the resolution of the underlying reanalysis.

Line 226. One should be cautious with referring to the origin of the rainfall using an analysis that is solely based on air parcel trajectories without taking into account the uptake of moisture along its pathways. The part of the sentence may be rephrased in the direction of "to generate a vector representing the mean trajectory of the air mass transport associated with the precipitation".

The titles of sections 4.1 and 4.2 may be rephrased as "Daily rainfall isotopic variability" and "Monthly rainfall isotopic variability".

Line 245. It may be helpful to refer to a study that presented the Global Meteoric Water Line. More importantly, a reader may expect after these two lines (244-247) an interpretation and discussion of the local meteoric water lines. What do we learn from the analysis? How do these local meteoric water lines compare to other regions? Later on, I realized that lines 281-285 further discuss this subject. The manuscript could benefit to describe this aspect at one place only (see also general comment 1).

Line 253. This synchronicity is quite remarkable as, according to this study, precipitation across the northern Iberian Peninsula is controlled by different weather regimes. May this suggest, along with later findings that show similar isotopic rainfall along the western and eastern coasts, that the elevation and temperature effects dominate the isotopic signatures in precipitation?

Lines 261-268. Here I miss again a discussion and interpretation of the results. Simply phrasing the main findings without interpretation leaves the reader guessing what to take away from the text. Later on, I realized that the text from line 286 onwards seems to continue with this analysis. Please, discuss one subjects at one place in the manuscript.

Line 315. In fact, when considering the above and following analysis, I get the impression that the elevation and/or temperature effect has the strongest influence on the rainfall isotopic variability (in the order of 2 permil) as compared to all other factors. Or is this too simplistic?

Lines 454-456. Another study that found similar differences in the isotopic signature in precipitation from convective versus stratiform precipitation in the Mediterranean is Lee et al. (2019). Citing this study may strengthen the text here.

Lines 460-462. The sentence "Backdoor cold fronts … … heavy precipitation and flooding (Llasat et al., 2007)" already appeared in section 2 (lines 107-109) and is thus repetitive. Please, remove the sentence at one of the two locations.

Lines 493-495. I cannot follow the sentence. Please, clarify and correct if necessary. In addition, how are outliers defined in Figure 7?

Tables. Overall, I find the information in the Tables quite overwhelming, and I wonder if the information can be reduced without losing relevant information. For example, the multiple use of "n=" in the cells of Table 2 could be avoid by choosing another notation, perhaps providing the number of samples between brackets after the $d^{18}O_p$ values, or simply by removing "n=" in all cells and providing adequate description on top of the columns or in the Table title/caption.

Lines 223-238. One of the main methodologies of the study is defining the different weather regimes that are linked to the rain events and $d^{18}O_p$ values. Upon first reading I missed how these different weather regimes are defined, and realized that lines 231-236 address this method. I would recommend to make this methodology more visible by renaming the title of section 3.4. In addition, more information should be provided on how these different synoptic situations are defined, allowing for potential reproduction of the results. Is this analysis subjective or based on an automated detection algorithm?

**Technical comments**

Line 32. Please, consider to remove the ";", and start a new sentence with "Atlantic …", and remove "studied".

Lines 37 and 38. Please, add a comma before "but" and consider to replace "plays a key control" by, for example, "plays a key role" or "exerts a key influence".

Lines 40, 44, 45 and elsewhere. Please, use a consistent phrasing; "palaeoclimate" or "paleoclimate".

Line 45. Please, remove the comma after "(e.g., Treble et al., 2005)".

Line 48. Please, remove the space between "history" and the comma, consider to rephrase the end of the sentence as "meteorological conditions, for example, air temperature and the amount of precipitation (Craig, 1961; Dansgaard, 1964)".

Line 50. Please, rephrase the end of the sentence as ", and the amount and source of the precipitation".

Line 52. The authors may consider to replace "mandatory if one wishes" by "essential".

Line 57. The authors may consider to rewrite "three major precipitation regimes" by "three major weather regimes of precipitation".

Line 63. The phrase "require further detailed and highly spatially-resolved studies." reads a little awkward. Please, consider rephrasing, for example, in the direction of "require further detailed studies that account for the high spatial variability of the regions and these processes.".

Line 67. Please, remove ", for example" and finish the sentence with a period.

Lines 68-69. Please, consider to replace "recovering" by "obtaining" or "covering" and write "at daily time scales".

Line 73. Please, rephrase "; basically, …" and start a new sentence with "Atlantic fronts are associated with more negative …"

Lines 81-82. Please, consider the rewrite the phrase "transect (850-km in straight line)", for example, by "transect of 850 km", or "transect of 850 km spherical distance", and the phrase "typical Atlantic climate to fully Mediterranean sites" by "typical Atlantic to a Mediterranean climate."

Line 96. Please, write "The Borrastre record".

Line 108. Please, write "they can cause".

Lines 113-114. Please, consider to replace "higher" by "enhanced" or "increased", and "influencing" by "affecting".

Lines 131-132. The sentence "Also located … … occurring mainly in spring and in autumn" reads awkward. Please, rephrase. It may help to start the sentence with "The Molinos site …".

Line 134. Please, consider to replace "is influenced by" by "has".

Line 148. Please, removed the word "Thus, "

Line 159. Please, replace "that the system was automatic" by, for example "that has an automatic system".

Line 163. I can't follow the logic of the sentence "Thus, … in 2015.". What does the sentence want to say? Please, rephrase, perhaps by just removing "Thus, ".

Line 168. Please, remove "daily" after "gauge".

Line 172. Please, consider to rephrase the sentence, for example as "…, are expressed as $d^{18}O$ and $d^2H$ in …".

Line 211. "at daily time scales" may read better.

Line 248. Please, write "daily time series".

Line 261. Please, write "All stations".

Line 283. Please, rephrase "is made with" as "is based on".

Line 286. Please, write "allows for an assessment".

Line 288. Please, rephrase, for example as "show quite similar monthly d18O values, and typically, more negative …".

Line 293 "…, is not what we expected." Sounds rather colloquial, please, rephrase.

Line 306. Please, consider to rephrase "to explain" as "that can explain"

Lines 320-321. I cannot follow the sentences "… and separated by the Mediterranean Sea. Therefore, the altitude cannot be …. difference between the studied sites.". Do the authors refer here to the difference in isotopic variability in the Alps and the northern Iberian Peninsula, or to the differences between the sites in the northern Iberian Peninsula? What is meant by "separated by the Mediterranean Sea"? Please, clarify and rephrase the text accordingly.

Line 324. Please, rephrase "air mass trajectory" by "air mass origin" or "moisture source region".

Line 328. Is the phrase "at daily and monthly time scales" in the section title needed? Please, consider to omit to shorten the title.

Line 331. Please, consider to replace "or" by "nor", and write "monthly time scales".

Lines 344 and 345. What does the notation "Oviedo+Pindal" and "Mallorca+Barcelona" mean?

Line 333. Please, replace "or" by "nor".

Line 360. It may be more adequate to speak about "origin of moisture for the rainfall".

Lines 374-376. This sentence reads awkward, please, rephrase.

Line 415. Please, consider to replace "more stable" by "similar" or "homogeneous" and add "stations" or "sites" after "the other two".

Line 424. Please, omit the word "whole".

Line 431. Please, omit the word "enter", and "it" at line 434.

Line 435 and elsewhere. Please, write out west and southwest.

Line 484. Please, consider to replace "circumstance" by "process".

Line 491. Please, write "backdoor frontal types".

Line 486. For clarity, the phrase "type of precipitation" may be complemented by ", i.e., convective versus frontal".

Lines 487-488. Please, consider to replace "in separating" by "on".

Line 503. Please, rephrase "the two opposite stations (the westernmost ones and the easternmost ones)", for example in the direction of "the two stations on opposite sides of the Iberian Peninsula" or "the two stations at the far most western and eastern sides of the Iberian Peninsula." or "the two stations at the western and eastern margins of the Iberian Peninsula.".

Line 719. Please, consider to replace "the remain ones" by "the remaining samples" and write "with respect to".

Line 725. The caption speaks about "Event"; I thought to understand from the text that the analysis is based on daily samples, not based on rain events?

Line 725-726. The current phrasing of the sentence is unclear. Please, correct, perhaps in the direction of "Note that El Pindal (2006-2009) samples are not represented since they do not overlap with the time period of the other stations."

Figure 4. I assume that the plots show the monthly data from January until December? Perhaps this can be included in the caption, if not in the Figure, for clarity.

Line 743. As a suggestion, "broken" may be replaced by "divided".

Lines 748-749. The colors and pressure levels in the caption are switched around! Red corresponds to 500 hPa and green to 850 hPa starting points. Please, correct.

Lines 768-769. Please, write "for every station" and "please see Table S2."

Line 776. The Table does not show numbers in red. Please, correct.

**References**

Dee, D. P., Uppala, S. M., Simmons, A. J., Berrisford, P., Poli, P., Kobayashi, S., Andrae, U., Balmaseda, M. A., Balsamo, G., Bauer, P., Bechtold, P., Beljaars, A. C. M., van de Berg, L., Bidlot, J., Bormann, N., Delsol, C., Dragani, R., Fuentes, M., Geer, A. J., Haimberger, L., Healy, S. B., Hersbach, H., Hólm, E. V., Isaksen, L., Kållberg, P., Köhler, M., Matricardi, M., McNally, A. P., Monge-Sanz, B. M., Morcrette, J.-J., Park, B.-K., Peubey, C., de Rosnay, P., Tavolato, C., Thépaut, J.-N., and Vitart, F.: The ERA-Interim reanalysis: configuration and performance of the data assimilation system, Q. J. Roy. Meteor. Soc., 137, 553–597, https://doi.org/10.1002/qj.828, 2011.

Lee, K.-O., Aemisegger, F., Pfahl, S., Flamant, C., Lacour, J.-L., and Chaboureau, J.-P.: Contrasting stable water isotope signals from convective and large-scale precipitation phases of a heavy precipitation event in southern Italy during HyMeX IOP 13: a modelling perspective, Atmos. Chem. Phys., 19, 7487–7506, https://doi.org/10.5194/acp-19-7487-2019, 2019

---

## Author Comment (AC1) · 19 Apr 2021

We have read Rev1's comments about our manuscript and appreciate his/her sincerity. We acknowledge his/her opinion about the novelty and interest of our data. However, we disagree with many of his/her observations as indicated in this response:

1) First, regarding the question of which factor influencing rainfall d18O composition is emphasized in this manuscript, we would like to note that the manuscript refers to a large number of factors and analyze the role they play in the variability of d18Orainfall. Our objective is to "assess the principal influencing factors determining rainfall isotopic

variability" although we agree we are not always able to quantify the effects of every factor since they are frequently playing an overlapping role. We state both in the abstract and in the conclusions the important role played by geographical factors when referring to annual averages and at a spatial approach but temperature and moisture origin and uptake are fundamental factors to explain the seasonality and the differences between "Atlantic" sites and "Mediterranean" ones. Therefore, it is fundamental to characterize the d18Orainfall at different sites and at different time scales. For example, in this manuscript we present for the first time in Spain the combination of seven sites to account for the regional spatial variability and the combination of daily and monthly data to account for the temporal scale. This is a huge exercise that for sure will be of interest for this community, even if we don't success on quantifying the effect of every factor separately.

2) Second, Rev1 considers the purpose of this study does not seem to fit the scope of the ACP based on the "local" character of our research. We totally disagree with this remark. In ACP there are many papers focused on a regional approach, not all the studies have general implications for atmospheric science as Rev1 indicates. Here is a short list of recent papers in ACP dealing with rainfall stable isotopes in quite local settings or focused on single events:

Bonne, J.-L., Masson-Delmotte, V., Cattani, O., Delmotte, M., Risi, C., Sodemann, H. and Steen-Larsen, H. C.: The isotopic composition of water vapour and precipitation in Ivittuut, southern Greenland, Atmospheric Chemistry and Physics, 14(9), 4419–4439, doi:https://doi.org/10.5194/acp-14-4419-2014, 2014.

Bonne, J.-L., Meyer, H., Behrens, M., Boike, J., Kipfstuhl, S., Rabe, B., Schmidt, T., Schönicke, L., Steen-Larsen, H. C. and Werner, M.: Moisture origin as a driver of temporal variabilities of the water vapour isotopic composition in the Lena River Delta, Siberia, Atmospheric Chemistry and Physics, 20(17), 10493–10511, doi:https://doi.org/10.5194/acp-20-10493-2020, 2020.

Dittmann, A., Schlosser, E., Masson-Delmotte, V., Powers, J. G., Manning, K. W., Werner, M. and Fujita, K.: Precipitation regime and stable isotopes at Dome Fuji, East Antarctica, Atmospheric Chemistry and Physics, 16(11), 6883–6900, doi:https://doi.org/10.5194/acp-16-6883-2016, 2016.

Jeelani, G., Deshpande, R. D., Galkowski, M. and Rozanski, K.: Isotopic composition of daily precipitation along the southern foothills of the Himalayas: impact of marine and continental sources of atmospheric moisture, Atmospheric Chemistry and Physics, 18(12), 8789–8805, doi:https://doi.org/10.5194/acp-18-8789-2018, 2018.

Lee, K.-O., Aemisegger, F., Pfahl, S., Flamant, C., Lacour, J.-L. and Chaboureau, J.-P.: Contrasting stable water isotope signals from convective and large-scale precipitation phases of a heavy precipitation event in southern Italy during HyMeX IOP 13: a modelling perspective, Atmospheric Chemistry and Physics, 19(11), 7487–7506, doi:https://doi.org/10.5194/acp-19-7487-2019, 2019.

Okazaki, A., Satoh, Y., Tremoy, G., Vimeux, F., Scheepmaker, R. and Yoshimura, K.: Interannual variability of isotopic composition in water vapor over western Africa and its relationship to ENSO, Atmospheric Chemistry and Physics, 15(6), 3193–3204, doi:https://doi.org/10.5194/acp-15-3193-2015, 2015.

Pfahl, S., Wernli, H. and Yoshimura, K.: The isotopic composition of precipitation from a winter storm – a case study with the limited-area model COSMOiso, Atmospheric Chemistry and Physics, 12(3), 1629–1648, doi:https://doi.org/10.5194/acp-12-1629-2012, 2012.

Steen-Larsen, H. C., Sveinbjörnsdottir, A. E., Peters, A. J., Masson-Delmotte, V., Guishard, M. P., Hsiao, G., Jouzel, J., Noone, D., Warren, J. K. and White, J. W. C.: Climatic controls on water vapor deuterium excess in the marine boundary layer of the North Atlantic based on 500 days of in situ, continuous measurements, Atmospheric Chemistry and Physics, 14(15), 7741–7756, doi:https://doi.org/10.5194/acp-14-7741-2014, 2014.

3) Finally, Rev1 propose presenting our manuscript as a Methodological report instead of a Research report. We agree this can be a good change and, after conversation to the Editor, we will move our manuscript to Methodological report format.

---

## Author Comment (AC2) · 19 Apr 2021

We appreciate very much the constructive comments by Rev2 which have certainly helped to improve our manuscript. Here we comment on some issues related to his/her notes which are of particular importance.

1) Shorten and strengthen the structure of the discussion on the climatic controls.

We totally agree with Rev2 ideas to shorten and strengthen the discussion and have mentioned the aspect of the difference in ocean surface isotope composition only once and shortly (now it is only included in section 5.3 and removed from the other lines

indicated by Rev2). Contrarily, we have dedicated more space to the controls at the moisture source following Rev2 advice, with new analyses of the moisture uptake (see below). Another idea to strengthen the structure of the discussion was to remove references to temperature control in p.16 and confine the discussion of temperature and precipitation only to Section 5.2. This is a good idea and, in fact, due to this and following Rev2 comment, we have decided to remove the discussion of temperature and amount of precipitation in relation to the type of rainfall. Thus, the last section of the discussion is only dedicated to changes in d18O according to rainfall types. We have modified Fig. 7 in this regard.

2) Temperature and precipitation amount controls on the variability of the isotopic composition of precipitation.

We agree with Rev2 about the relative role that temperature may exert on our data and about the difficulty to explain this role without fully understanding the implied processes and all the overlapping interactions. Therefore, we have modified this section to modulate our results (eg. indicating the correlation numbers without implying causality). We are aware that behind the "temperature effect" there are many other processes and mechanisms not easy to describe just by employing correlation analyses. This idea is again employed in Section 5 introductory paragraph, as Rev2 suggested.

3) Moisture source identification discussion.

We greatly appreciate Rev2 suggestions on this topic and have included a new analysis to calculate moisture uptake in all events (850hpa trajectories). This idea was also expressed by Rev 3 and it is possibly the most important change we have made in this new version. We use Baldini's method (Baldini et al., 2010) in a more restrictive way (see also Iglesias González, 2019) to identify the locations where moisture uptake processes have been produced during the 48h before the rainfall samples were collected. Taking into account that Iberian Peninsula is surrounded by ocean, together with the fact that most of the rainfall events analyzed in the investigation were produced Interactive comment

by frontal systems and convection events (see synoptic analysis), only 850hPa airmass moisture uptake events have been considered as relevant in our new analysis. In addition, while Baldini et al, (2010) considered moisture uptake processes with an increase in 1h of 0.1 gH2Ov/kgair as significant, in our analysis we only took into account events where moisture uptake process where higher than 0.25 gH2Ov/kgair, so if exists any influence in the rainfall isotopical signal, it would be easier to identify than in other previous studies. With this restricted method, and considering all the events analyzed, more than 3000 moisture uptake events have been identified. These events were analyzed considering seasonal variability and the different locations where the rainfall samples were collected. With this new analysis, we are able to identified changes in the moisture uptake location distribution of the airmasses which produces rainfall events along the lberian transect. These results are now discussed in detail and represented in a new figure.

4) Seasonality of moisture sources

We mention now in the introduction the role of seasonality of moisture sources. This idea is later revised in the discussion (section 5.3) where it becomes clear after the moisture origin and uptake study.

5) Meteorological context of precipitation events

We appreciate the suggestions made by Rev2 about other studies related to the isotopic composition of different precipitation types. Most of those references are now included in the text. However, Rüdisühli et al. 2020 is included in section 2 when describing which weather system dominates precipitation in which season on the Iberian Peninsula.

**Minor comments**

We include here our responses to other (minor) comments from Rev2 (those not included here in the letter were minor comments and were just corrected following re-
viewer's suggestions).

- Rev2 considers the abstract is not well structured and not totally clear in the factors we propose as the most important ones. We agree about the fact that all investigated processes are overlapping and we just show a consistent picture from different angles. We have used this idea in the abstract (also it appears at the beginning of the Discussion section).

- Regarding our text about the NAO, we agree with Rev2 who considered it was out of scope since it was not later used in the analyses performed. Thus, we have removed it from this revised version.

- We agree about the importance of the sampling procedure and have highlighted that only at one site (from the 7 sites) we used an automated system (at El Pindal site). More details on that self-built system are now presented. Unfortunately, at the time of the experiment (2006-2008), we have not compared our system with rainfall collected by other automated or manually procedures.

- According to Rev2 comment, we have moved to the method section lines 392-399 in page 13 and better described our method to calculate and represent back trajectories.

- In line with Rev2 about the reason to use disaggregated precipitation time series, we want to remark we follow the procedure carried out by Millan et al. (2005) to account for the meteorological origin of every rainfall event. We decided to apply this approach to analyze differences between isotopes as a function of three moisture source regions, i.e.: (i) Atlantic frontal systems; (ii) convective-orographic storms; and (iii) easterly advections over the Mediterranean sea (back-door cold fronts). It is well-known in Meteorology that the atmospheric dynamics and evaporation behind precipitation from these three components are different. Thus, this procedure allowed us to discern one of the principal influencing factors (type of precipitation) determining rainfall isotopic composition and variability. For instance, we found higher isotopic measurements associated with convective-orographic storms. This explanation was already included in the text,

ACPD
in Methods, but slightly improved in this new version.

- We have prepared new figures following Rev2 suggestion of splitting Fig 3 into different panels (one per station). We add to every site, the deuterium excess and total precipitation. We include those figures in the supplementary material since we agree they are important but not the focus of this manuscript.

- Rev2 emphasizes the large daily variability of our dataset and this idea is now included in the text. Similarly, we have include a sentence in the short introductory paragraph of the Discussion section making reference to the high complexity of the hydrological cycle with many processes playing a role (or a combination of processes interacting) in the formation of the rainfall isotopic signal.

- In discussion section 5.1, Rev2 points to two sentences or paragraphs that should be moved to section 5.3 and section 5.2. We understand his/her reasons, and have modified these sections to avoid repetitions. Still, the interaction among processes and drivers make inadequate a rigid organization in the discussion and, inevitably, some effects are already introduced in 5.1.

- As suggested by Rev2, we incorporate more often the classification of our studied sites into subregions (Cantabrian coast, Iberian range, Pyrenees and Mediterranean) in the paper (already used to describe the sites, section 2).

- Rev2 indicates that we don't show the important role played by the moisture source in a quantitative and methodologically convincing way. We don't totally agree with this criticism, specially since in Figure 5, the role of moisture source, underlined by the air masses trajectories, is quantified (in percentages of rainfall and in isotopic values). To us, it is evident in Fig.5 the two dominant sources in Borrastre sites, as indicated in the text. In Table 4, the three main synoptic patterns are also quantified for the seven studied sites. In any case, the moisture uptake study carried out in this new version represents a more convincing way of representing the role of the air masses origin.
- We have updated our conclusions following main changes carried out in the new revised version. However, we are not including more discussion on the manuscript about d-excess to not complicate it. A previous version of this study incorporated d-excess data and we found too complicated for us to integrate it in the discussion. We prefer to make those data available (d-excess is represented for every site in the new supplementary figure and data are available at the supplementary tables) giving the opportunity to other experts on that subject to use this large dataset.

Cited references

Baldini, L. M., McDermott, F., Baldini, J. U. L., Fischer, M. J., and Möllhoff, M.: An investigation of the controls on Irish precipitation  $\delta$ 18O values on monthly and event timescales, Clim Dyn, 35, 977–993, https://doi.org/10.1007/s00382-010-0774-6, 2010.

Iglesias González, M. I.: Variabilidad climática del noroeste de la península ibérica durante los últimos 1500 años, descrita por espeleotemas de diversas cuevas del principado de asturias, http://purl.org/dc/dcmitype/Text, Universidad de Oviedo, 2019.

**ACPD**

---

## Author Comment (AC3) · 19 Apr 2021

We appreciate very much the constructive comments by Rev3 which have certainly helped to improve our manuscript. The level of detail of his/her comments is extraordinary and very helpful. Here we comment on some issues related to his/her notes which are of particular importance.

General comments

1) Organization of the manuscript and writing. We appreciate the recommendations from Rev3 to improve the structure of the manuscript (eg. merging chapters 4 and 5)

and, accordingly, have modified the organization ending with a Results and Discussion section divided in six subsections. We also avoid repetitions, such as the two previous sections in the text where we talk about meteoric water lines. Some of these ideas are in line with those proposed by Rev2 to shorten the manuscript, so we think the final version is certainly improved and have increased its readability.

2) Source regions and backward trajectories. Rev3 considers insufficient our study of back trajectories to discriminate the moisture source at the study transect. We agree with this argument and it probably represents the largest change we have carried out in this version. In fact, it is true, that most trajectories have an origin in the NW, but they later follow a sometimes quite complicated path with different options of moisture uptake. Therefore, we agree that the study of the trajectories alone is not able to represent the processes we want. Therefore we, first, have replaced Fig. 5 by S1 as Rev3 suggested to obtain our results from trajectories extracted the last 1 or 2 days. Second, we have performed a new analysis to calculate moisture uptake in all events (850hpa trajectories). We use Baldini's method (Baldini et al., 2010) in a more restrictive way (see also Iglesias González, 2019) to identify the locations where moisture uptake processes have been produced during the 48h before the rainfall samples were collected. Taking into account that Iberian Peninsula is surrounded by ocean, together with the fact that most of the rainfall events analyzed in the investigation were produced by frontal systems and convection events (see synoptic analysis), only 850hPa airmass moisture uptake events have been considered as relevant in our new analysis. In addition, while Baldini et al, (2010) considered moisture uptake processes with an increase in 1h of 0.1 gH2Ov/kgair as significant, in our analysis we only took into account events where moisture uptake process where higher than 0.25 gH2Ov/kgair, so if exists any influence in the rainfall isotopical signal, it would be easier to identify than in other previous studies. With this restricted method, and considering all the events analyzed, more than 3000 moisture uptake events have been identified. These events were analyzed considering seasonal variability and the different locations where the rainfall samples were collected. With this new analysis, we are able to identified changes in the mois-

ture uptake location distribution of the airmasses which produces rainfall events along the Iberian transect. These results are now discussed in detail and represented in a new figure.

Specific comments (Rev3 comments are indicated as RV3)

RV3 Line 2. The title refers to "climate controls" on the variability of isotopic composition in rainfall. The study itself though seems more to be focused on meteorological processes such as moisture pathways and weather regimes/precipitation types of rain days. Perhaps, the authors may consider to use or add another term such as "weather", "meteorological", or "atmospheric"?

Good suggestion! We use atmospheric.

RV3 Line 39-40. Perhaps, besides referring to the dataset, this concluding sentence may also refer to the analysis that helps to understand rainfall isotope variability in relation to meteorological / atmospheric processes and geographic influences?

Good suggestion! Change "dataset" by "analyses".

RV3 Lines 73 and 74. The term "trajectories" is perhaps quite technical for the introduction. Instead, a term that refers to actual physical processes, such as "air mass origins" or "air mass transport" may be more appropriate.

Done

RV3 Lines 80-85. This is a crucial paragraph as it outlines what the intention of the study is, and what it adds to previous studies as outlined in the text above. The thought behind the first sentence "In this paper . . ." is not clear to me. Is the approach, based on multiple stations new and is that the main selling point of the paper? Or is this study presenting a comprehensive analysis based on multiple stations across the Atlantic-Mediterranean transect? In the first case, the authors may write "we introduce a new approach. . .", and in the latter case, "we present a comprehensive / multiple perspective analysis on daily and monthly . . .". Also, is it really new that a study considers

multiple stations across a region? If other studies followed such an approach, perhaps for other regions, this may deserve attention in the introduction to provide context for this study, for example by adding a new paragraph. In addition, this paragraph may explicitly refer to the atmospheric processes and geographic factors that influence the isotopic rainfall variability that are addressed in this study to guide the reader's expectations.

We are presenting a comprehensive analyses based on multiple stations, not certainly "a new approach" since there are many studies using multiple stations. Some of those previous references are included now in the introduction. We also added some information about the processes and factors we are going to address in the manuscript.

RV3 Line 87. This section addresses besides the site description and climate also the different weather regimes that bring precipitation over the northern Iberian Peninsula. This may be reflected in the title of the section.

Done.

RV3 Lines 103-104. The phrase "also easterly advections over the Mediterrean Sea" sounds somewhat vague. Please, rewrite, perhaps in the direction of "fronts that approach the Iberian Peninsula from the east (backdoor cold fronts)"..

Done

RV3 Lines 119-122. While reading this paragraph I somehow lost the storyline. The first sentence refers to the dominant source regions and seems to follow as a conclusion from the text above, while the next sentence introduces the four different climate zones. The authors may consider to add the first sentence to the paragraph above (or elsewhere), and to start a new paragraph with the second sentence.

Yes, we agree and have removed the first sentence.

RV3 Then, the introduction of the four climate zone regions is hard to follow; It may help rephrase this sentence as, for example, "Below, the seven stations are grouped

into four regions and described in terms of their climatology". Also, it feels somewhat chaotic to refer at this stage multiple times to Figure 4 while Figures 2 and 3 have not yet been discussed. Is it necessary to include the line "Regional meteorological data are provided in Figure 4A."?

Done. We have removed references to Fig. 4 that were unnecessary.

RV3 Lines 123-127. Can this paragraph be shortened by saying "The sites of El Pindal and Oviedo..." and removing the sentence on lines 126-127 "Additionally, ... in this study."?

Done

RV3 Line 197. To what "Meteorological data" is referred? If this is the air temperature and precipitation, please, remove the brackets, and rephrase the sentence to place more emphasis on these meteorological variables, for example, as "Air temperature and precipitation are obtained from the closest meteorological stations over the sampling periods, as indicated in Table 1, to investigate .... "

Done

RV3 Line 292. Usually, when referring to the ERA-Interim analysis Dee et al. (2011) is cited.

Done.

RV3 Lines 211-238. In this paragraph I feel quite overwhelmed by the many references to Tables and Figures for which here only the applied methodology is described (e.g., Tables 3, 4, and 5 and Figure 5). I would recommend to only refer explicitly to the Tables and Figures when discussing the scientific results, not when describing the used (statistical) methods.

We partially agree about this... but citing tables here is quite necessary to refer to the place where the reader can find the data associated to that analysis. We have kept the

references to Tables and removed those to Figures.

RV3 Lines 223-224. Which reanalysis data are the HYSPLIT simulations using? This should briefly be mentioned, including the resolution of the underlying reanalysis.

Done. We have included this brief sentence: "GDAS (Global Data Asimilation System) have been used in Hysplit simulations with $0.5°x0.5°$ spatial resolution".

RV3 Line 226. One should be cautious with referring to the origin of the rainfall using an analysis that is solely based on air parcel trajectories without taking into account the uptake of moisture along its pathways. The part of the sentence may be rephrased in the direction of "to generate a vector representing the mean trajectory of the air mass transport associated with the precipitation".

Yes, we agree. In this new version, a procedure to consider moisture uptake is included (see general comments above)

RV3 The titles of sections 4.1 and 4.2 may be rephrased as "Daily rainfall isotopic variability" and "Monthly rainfall isotopic variability".

Done

RV3 Line 245. It may be helpful to refer to a study that presented the Global Meteoric Water Line. More importantly, a reader may expect after these two lines (244-247) an interpretation and discussion of the local meteoric water lines. What do we learn from the analysis? How do these local meteoric water lines compare to other regions? Later on, I realized that lines 281-285 further discuss this subject. The manuscript could benefit to describe this aspect at one place only (see also general comment 1).

We have better organized this section and merged the two places where the meteoric water lines were described. Comparison with other sites in southern France is now included.

RV3 Line 253. This synchronicity is quite remarkable as, according to this study, precipitation across the northern Iberian Peninsula is controlled by different weather regimes. May this suggest, along with later findings that show similar isotopic rainfall along the western and eastern coasts, that the elevation and temperature effects dominate the isotopic signatures in precipitation?

We think that this case is quite singular since it represents the influence of an Atlantic front passing over a large region of the IP and affecting our sites in a similar way (high precipitation amount, very negative isotopes). It may be difficult to extrapolate this quite exceptional situation to the whole record and extract general conclusions.

RV3 Lines 261-268. Here I miss again a discussion and interpretation of the results. Simply phrasing the main findings without interpretation leaves the reader guessing what to take away from the text. Later on, I realized that the text from line 286 onwards seems to continue with this analysis. Please, discuss one subjects at one place in the manuscript.

This text was just a presentation of the data since this section was in Result chapter in the previous version of the manuscript. Now we have included the discussion of the data, adding information previously on line 286 onwards.

RV3 Line 315. In fact, when considering the above and following analysis, I get the impression that the elevation and/or temperature effect has the strongest influence on the rainfall isotopic variability (in the order of 2 permil) as compared to all other factors. Or is this too simplistic?

It is true that elevation and temperature are important to explain averaged values (eg. annual means) but not enough to explain daily variability. For that scale, we need to account for the air mass history (moisture origin and moisture uptake, type of rainfall, etc).

RV3 Lines 454-456. Another study that found similar differences in the isotopic signature in precipitation from convective versus stratiform precipitation in the Mediterranean

is Lee et al. (2019). Citing this study may strengthen the text here.

Yes, this paragraph is enriched with new references provided by both Rev2 and Rev3.

RV3 Lines 460-462. The sentence "Backdoor cold fronts . . . . . . heavy precipitation and flooding (Llasat et al., 2007)" already appeared in section 2 (lines 107-109) and is thus repetitive. Please, remove the sentence at one of the two locations.

Done. We remove it from the discussion (section 4.6).

RV3 Lines 493-495. I cannot follow the sentence. Please, clarify and correct if necessary. In addition, how are outliers defined in Figure 7?

Following recommendation by Rev2, this last paragraph associating rainfall types with precipitation amount or temperature has been removed since it was rather speculative. We have also simplified Fig. 7 to show only the variation of d18O associated to the three rainfall types

RV3 Tables. Overall, I find the information in the Tables quite overwhelming, and I wonder if the information can be reduced without losing relevant information. For example, the multiple use of "n=" in the cells of Table 2 could be avoid by choosing another notation, perhaps providing the number of samples between brackets after the d18Op values, or simply by removing "n=" in all cells and providing adequate description on top of the columns or in the Table title/caption.

Done. Table 2 is simplified according to these suggestions.

RV3 Lines 223-238. One of the main methodologies of the study is defining the different weather regimes that are linked to the rain events and d18Op values. Upon first reading I missed how these different weather regimes are defined, and realized that lines 231-236 address this method. I would recommend to make this methodology more visible by renaming the title of section 3.4. In addition, more information should be provided on how these different synoptic situations are defined, allowing for potential reproduction of the results. Is this analysis subjective or based on an automated

detection algorithm?

We have changed the title in the method section to make this methodology more visible and provide more information about where to find the criteria (subjective) to define the three different synoptic situations.

We have also changed all the typos and other errors indicated by Rev3 in "Technical comments" thus improving this new version of the manuscript.

Cited references

Baldini, L. M., McDermott, F., Baldini, J. U. L., Fischer, M. J., and Möllhoff, M.: An investigation of the controls on Irish precipitation $\delta$18O values on monthly and event timescales, Clim Dyn, 35, 977–993, https://doi.org/10.1007/s00382-010-0774-6, 2010.

Iglesias González, M. I.: Variabilidad climática del noroeste de la península ibérica durante los últimos 1500 años, descrita por espeleotemas de diversas cuevas del principado de asturias, http://purl.org/dc/dcmitype/Text, Universidad de Oviedo, 2019.